# Implicit Neural Representations with Periodic Activation Functions

**Vincent Sitzmann**\*
sitzmann@cs.stanford.edu

**Julien N. P. Martel**\*
jnmartel@stanford.edu

**Alexander W. Bergman**
awb@stanford.edu

**David B. Lindell**
lindell@stanford.edu

**Gordon Wetzstein**
gordon.wetzstein@stanford.edu

Stanford University
vsitzmann.github.io/siren/

## Abstract

Implicitly defined, continuous, differentiable signal representations parameterized by neural networks have emerged as a powerful paradigm, offering many possible benefits over conventional representations. However, current network architectures for such implicit neural representations are incapable of modeling signals with fine detail. They also fail to accurately model spatial and temporal derivatives, which is necessary to represent signals defined implicitly by differential equations. We propose to leverage periodic activation functions for implicit neural representations and demonstrate that these networks, dubbed sinusoidal representation networks or SIRENs, are ideally suited for representing complex natural signals and their derivatives. We analyze SIREN activation statistics to propose a principled initialization scheme and demonstrate the representation of images, wavefields, video, sound, three-dimensional shapes, and their derivatives. Further, we show how SIRENs can be leveraged to solve challenging boundary value problems, such as particular Eikonal equations (yielding signed distance functions), the Poisson equation, and the Helmholtz and wave equations. Lastly, we combine SIRENs with hypernetworks to learn priors over the space of SIREN functions. Please see the project website for a video overview of the proposed method and all applications.

## 1 Introduction

We are interested in a class of functions $\Phi$ that satisfy equations of the form:

$$\mathcal{C}\left(\mathbf{x}, \Phi, \nabla_{\mathbf{x}}\Phi, \nabla_{\mathbf{x}}^2\Phi, \ldots\right) = 0, \quad \Phi : \mathbf{x} \mapsto \Phi(\mathbf{x}). \quad (1)$$

In this implicit problem formulation, a functional $\mathcal{C}$ takes as input the spatial or spatio-temporal coordinates $\mathbf{x} \in \mathbb{R}^m$ and, optionally, derivatives of $\Phi$ with respect to these coordinates. Our goal is then to learn a neural network that parameterizes $\Phi$ to map $\mathbf{x}$ to some quantity of interest while satisfying the constraint presented in Equation (1). Thus, $\Phi$ is implicitly defined by the relation modeled by $\mathcal{C}$ and we refer to neural networks that parameterize such implicitly defined functions as *implicit neural representations*. As we show in this paper, a surprisingly wide variety of problems across scientific fields fall into this form, such as modeling many different types of discrete signals in image, video, and audio processing using a continuous and differentiable representation, learning 3D shape representations via signed distance functions [1–4], and, more generally, solving boundary value problems, such as the Poisson, Helmholtz, or wave equations.

---

A continuous parameterization offers several benefits over alternatives, such as discrete grid-based representations. For example, due to the fact that $\Phi$ is defined on the continuous domain of $\mathbf{x}$, it can be significantly more memory efficient than a discrete representation, allowing it to model fine detail that is not limited by the grid resolution but by the capacity of the underlying network architecture. As an example, we show how our SIREN architecture can represent complex 3D shapes with networks using only a few hundred kilobytes whereas naive mesh representations of the same datasets require hundreds of megabytes. Being differentiable implies that gradients and higher-order derivatives can be computed analytically, for example using automatic differentiation, which again makes these models independent of conventional grid resolutions. Finally, with well-behaved derivatives, implicit neural representations may offer a new toolbox for solving inverse problems, such as differential equations.

For these reasons, implicit neural representations have seen significant research interest over the last year (Sec. 2). Most of these recent representations build on ReLU-based multilayer perceptrons (MLPs). While promising, these architectures lack the capacity to represent fine details in the underlying signals, and they typically do not represent the derivatives of a target signal well. This is partly due to the fact that ReLU networks are piecewise linear, their second derivative is zero everywhere, and they are thus incapable of modeling information contained in higher-order derivatives of natural signals. While alternative activations, such as tanh or softplus, are capable of representing higher-order derivatives, we demonstrate that their derivatives are often not well behaved and also fail to represent fine details.

To address these limitations, we leverage MLPs with periodic activation functions for implicit neural representations. We demonstrate that this approach is not only capable of representing details in the signals better than ReLU-MLPs, or positional encoding strategies proposed in concurrent work [5], but that these properties also uniquely apply to the derivatives, which is critical for many applications we explore in this paper.

To summarize, the contributions of our work include:

- A continuous implicit neural representation using periodic activation functions that fits complicated signals, such as natural images and 3D shapes, and their derivatives robustly.

- An initialization scheme for training these representations and validation that distributions of these representations can be learned using hypernetworks.

- Demonstration of applications in image, video, and audio representation; 3D shape reconstruction; solving first-order differential equations to estimate a signal from its gradients; and solving second-order differential equations.

## 2   Related Work

**Implicit neural representations.**   Recent work has demonstrated the potential of fully connected networks as continuous, memory-efficient implicit representations for shape parts [6, 7], objects [1, 4, 8, 9], or scenes [10–12]. These representations are typically trained from some form of 3D data as either signed distance functions [1, 4, 8–12] or occupancy networks [2, 13]. In addition to representing shape, some of these models have been extended to also encode object appearance [3, 5, 10, 14, 15], which can be trained using (multiview) 2D image data using neural rendering [16]. Temporally aware extensions [17] and variants that add part-level semantic segmentation [18] have also been proposed.

**Periodic nonlinearities.**   Periodic nonlinearities have been investigated repeatedly over the past decades, but have so far failed to robustly outperform alternative activation functions. Early work includes Fourier neural networks, engineered to mimic the Fourier transform via single-hidden-layer networks [19, 20]. Other work explores neural networks with periodic activations for simple classification tasks [21–23], equation learning [24], and recurrent neural networks [25–29]. For such models, the training dynamics have been investigated [30], and it has been shown that they have universal function approximation properties [31–33]. Compositional pattern producing networks [34, 35] also leverage periodic nonlinearities, but rely on a combination of different nonlinearities via evolution in a genetic algorithm framework. Motivated by the discrete cosine transform, Klocek et al. [36] leverage cosine activation functions for image representation but they do not study the derivatives of these representations or other applications explored in our work. Inspired by these and

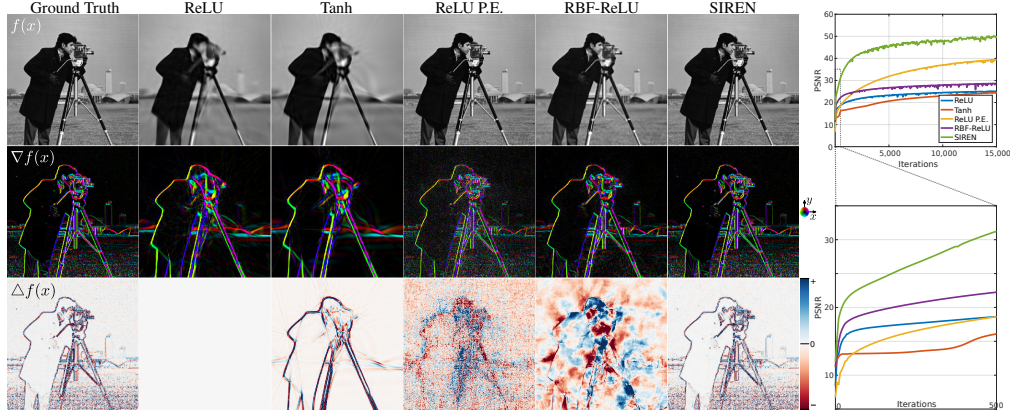

Figure 1: Comparison of different neural network architectures fitting the implicit representation of an image (ground truth: top left). The representation is only supervised on the target image but we also show first- and second-order derivatives of the function fit in rows 2 and 3, respectively.

other seminal works, we explore MLPs with periodic activation functions for applications involving implicit neural representations and their derivatives, and we propose principled initialization and generalization schemes.

**Neural DE Solvers.** Neural networks have long been investigated in the context of solving differential equations (DEs) [37], and have previously been introduced as implicit representations for this task [38]. Early work on this topic involved simple neural network models, consisting of MLPs or radial basis function networks with few hidden layers and hyperbolic tangent or sigmoid nonlinearities [38–41]. The limited capacity of these shallow networks typically constrained results to 1D solutions or simple 2D surfaces. Modern approaches to these techniques leverage recent optimization frameworks and auto-differentiation, but use similar architectures based on MLPs. Still, solving more sophisticated equations with higher dimensionality, more constraints, or more complex geometries is feasible [42–45]. However, we show that the commonly used MLPs with smooth, non-periodic activation functions fail to accurately model high-frequency information and higher-order derivatives even with dense supervision.

Neural ODEs [46] are related to this topic, but are very different in nature. Whereas implicit neural representations can be used to directly solve ODEs or PDEs from supervision on the system dynamics, neural ODEs allow for continuous function modeling by pairing a conventional ODE solver (e.g., implicit Adams or Runge-Kutta) with a network that parameterizes the dynamics of a function. The proposed architecture may be complementary to this line of work.

## 3 Formulation

Our goal is to solve problems of the form presented in Equation (1). We cast this as a feasibility problem, where a function $\Phi$ is sought that satisfies a set of $M$ constraints $\{\mathcal{C}_m(\mathbf{a}(\mathbf{x}), \Phi(\mathbf{x}), \nabla\Phi(\mathbf{x}), ...)\}_{m=1}^M$, each of which relate the function $\Phi$ and/or its derivatives to quantities $\mathbf{a}(\mathbf{x})$:

$$\text{find } \Phi \text{ subject to } \mathcal{C}_m\big(\mathbf{a}(\mathbf{x}), \Phi(\mathbf{x}), \nabla\Phi(\mathbf{x}), ...\big) = 0, \ \forall\mathbf{x} \in \Omega_m, \ m = 1, \ldots, M \quad (2)$$

This problem can be cast in a loss function that penalizes deviations from each of the constraints on their domain $\Omega_m$:

$$\mathcal{L} = \int_\Omega \sum_{m=1}^M \mathbf{1}_{\Omega_m}(\mathbf{x}) \left\| \mathcal{C}_m(\mathbf{a}(\mathbf{x}), \Phi(\mathbf{x}), \nabla\Phi(\mathbf{x}), ...) \right\| d\mathbf{x}, \quad (3)$$

with the indicator function $\mathbf{1}_{\Omega_m}(\mathbf{x}) = 1$ when $\mathbf{x} \in \Omega_m$ and 0 when $\mathbf{x} \notin \Omega_m$. In practice, the loss function is enforced by sampling $\Omega$. A dataset $\mathcal{D} = \{(\mathbf{x}_i, \mathbf{a}_i(\mathbf{x}))\}_i$ is a set of tuples of coordinates $\mathbf{x}_i \in \Omega$ along with samples from the quantities $\mathbf{a}(\mathbf{x}_i)$ that appear in the constraints. Thus, the loss in Equation (3) is enforced on coordinates $\mathbf{x}_i$ sampled from the dataset, yielding the loss

$\tilde{\mathcal{L}} = \sum_{i \in \mathcal{D}} \sum_{m=1}^{M} \mathbf{1}_{\Omega_m}(\mathbf{x}) \left\| \mathcal{C}_m(\mathbf{a}(\mathbf{x}_i), \Phi(\mathbf{x}_i), \nabla\Phi(\mathbf{x}_i), ...) \right\|$. In practice, the dataset $\mathcal{D}$ is sampled dynamically at training time, approximating $\mathcal{L}$ better as the number of samples grows, as in Monte Carlo integration.

We parameterize functions $\Phi_\theta$ as fully connected neural networks with parameters $\theta$, and solve the resulting optimization problem using gradient descent. The derivatives in Eq. (3) such as $\nabla_{\mathbf{x}}\Phi_\theta$ correspond to the gradient of the network's outputs with respect to its inputs. Those can be computed with auto-differentiation [47]. They are automatically added to the computation graph when defining the loss function, thus enabling the optimization of the weights $\theta$ during training.

## 3.1 Periodic Activations for Implicit Neural Representations

We propose SIREN, a simple neural network architecture for implicit neural representations that uses the sine as a periodic activation function:

$$\Phi(\mathbf{x}) = \mathbf{W}_n \left( \phi_{n-1} \circ \phi_{n-2} \circ \ldots \circ \phi_0 \right)(\mathbf{x}) + \mathbf{b}_n, \quad \phi_i(\mathbf{x}_i) = \sin\left( \mathbf{W}_i \mathbf{x}_i + \mathbf{b}_i \right). \tag{4}$$

Here, $\phi_i : \mathbb{R}^{M_i} \mapsto \mathbb{R}^{N_i}$ is the $i^{th}$ layer of the network. It consists of the affine transform defined by the weight matrix $\mathbf{W}_i \in \mathbb{R}^{N_i \times M_i}$ and the biases $\mathbf{b}_i \in \mathbb{R}^{N_i}$ applied on the input $\mathbf{x}_i \in \mathbb{R}^{M_i}$, followed by the sine nonlinearity applied to each component of the resulting vector.

Interestingly, any derivative of a SIREN *is itself a composition of* SIREN*s*, as the derivative of the sine is a cosine, i.e., a phase-shifted sine (see supplemental). Therefore, the derivatives of a SIREN inherit the properties of SIRENs, enabling us to supervise any derivative of SIREN with "complicated" signals. In our experiments, we demonstrate that when a SIREN is supervised using a constraint $\mathcal{C}_m$ involving the derivatives of $\Phi$, the function realized by the neural network $\Phi_\theta$ remains well behaved, which is crucial in solving many problems, including boundary value problems (BVPs). In contrast to conventional nonlinearities such as the hyperbolic tangent or the ReLU, the sine is periodic and therefore, non-local. Intuitively, this provides SIREN with a degree of shift invariance, as it may learn to apply the same function to different input coordinates.

We will show that SIRENs can be initialized with some control over the distribution of activations, allowing us to create deep architectures. Furthermore, SIRENs converge significantly faster than baseline architectures, fitting, for instance, a single image in a few hundred iterations, taking a few seconds on a modern GPU, while featuring higher image fidelity (Fig. 1).

**A simple example: fitting an image.** Consider the case of finding the function $\Phi : \mathbb{R}^2 \mapsto \mathbb{R}^3$ that parameterizes a given discrete image $f$ in a continuous fashion. The image defines a dataset $\mathcal{D} = \{(\mathbf{x}_i, f(\mathbf{x}_i))\}_i$ of pixel coordinates $\mathbf{x}_i = (x_i, y_i)$ associated with their RGB colors $f(\mathbf{x}_i)$. The only constraint $\mathcal{C}$ is that $\Phi$ should output image colors at pixel coordinates. $\mathcal{C}$ solely depends on both $\Phi$ (none of its derivatives) and $f(\mathbf{x}_i)$, with the form: $\mathcal{C}(f(\mathbf{x}_i), \Phi(\mathbf{x})) = \Phi(\mathbf{x}_i) - f(\mathbf{x}_i)$ which can be translated into the loss $\tilde{\mathcal{L}} = \sum_i \|\Phi(\mathbf{x}_i) - f(\mathbf{x}_i)\|$. In Fig. 1, we fit $\Phi_\theta$ using comparable

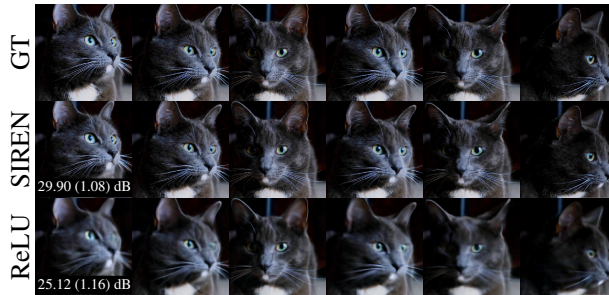

Figure 2: Example frames from fitting a video with SIREN and ReLU-MLPs. Our approach faithfully reconstructs fine details like the whiskers. Mean (and standard deviation) of the PSNR over all frames is reported.

network architectures with different activation functions to a natural image. We supervise this experiment only on the image values, but also visualize the gradients $\nabla\Phi$ and Laplacians $\Delta\Phi$. While only two approaches, a ReLU network with positional encoding (P.E.) [5, 48] and our SIREN, accurately represent the ground truth image $f(\mathbf{x})$, SIREN is the only network capable of also representing the derivatives of the signal. Additionally, we run a simple experiment where we fit a short video with 300 frames and with a resolution of 512×512 pixels using both ReLU and SIREN MLPs. As seen in Figure 2, our approach is successful in representing this video with an average peak signal-to-noise ratio close to 30 dB, outperforming the ReLU baseline by about 5 dB. We also show the flexibility of SIRENs by representing audio signals in the supplement.

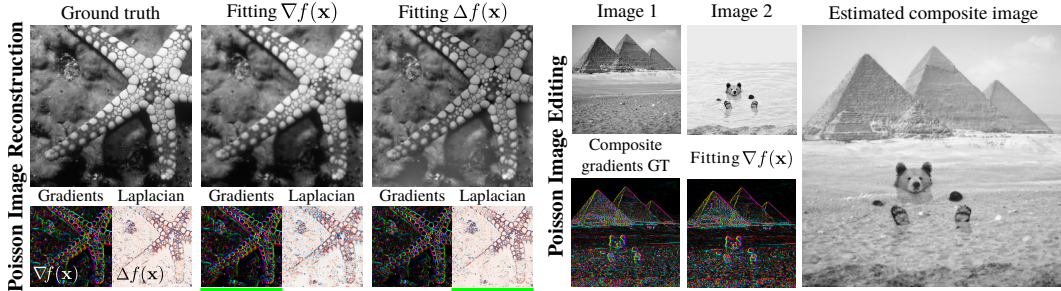

Figure 3: **Poisson image reconstruction:** An image (left) is reconstructed by fitting a SIREN, supervised either by its gradients or Laplacians (underlined in green). The results, shown in the center and right, respectively, match both the image and its derivatives well. **Poisson image editing:** The gradients of two images (top) are fused (bottom left). SIREN allows for the composite (right) to be reconstructed using supervision on the gradients (bottom right).

## 3.2   Distribution of activations, frequencies, and a principled initialization scheme

We present a principled initialization scheme necessary for the effective training of SIRENs. While presented informally here, we discuss further details, proofs and empirical validation in the supplemental material. The key idea in our initialization scheme is to preserve the distribution of activations through the network so that the final output at initialization does not depend on the number of layers. Note that building SIRENs without carefully chosen weights yielded poor performance both in accuracy and in convergence speed.

To this end, let us first consider the output distribution of a single sine neuron with the uniformly distributed input $x \sim \mathcal{U}(-1, 1)$. The neuron's output is $y = \sin(ax + b)$ with $a, b \in \mathbb{R}$. It can be shown that for any $a > \frac{\pi}{2}$, i.e. spanning at least half a period, the output of the sine is $y \sim \arcsin(-1, 1)$, a special case of a U-shaped Beta distribution and independent of the choice of $b$. We can now reason about the output distribution of a neuron. Taking the linear combination of $n$ inputs $\mathbf{x} \in \mathbb{R}^n$ weighted by $\mathbf{w} \in \mathbb{R}^n$, its output is $y = \sin(\mathbf{w}^T\mathbf{x} + b)$. Assuming this neuron is in the second layer, each of its inputs is arcsine distributed. When each component of $\mathbf{w}$ is uniformly distributed such as $w_i \sim \mathcal{U}(-c/\sqrt{n}, c/\sqrt{n}), c \in \mathbb{R}$, we show (see supplemental) that the dot product converges to the normal distribution $\mathbf{w}^T\mathbf{x} \sim \mathcal{N}(0, c^2/6)$ as $n$ grows. Finally, feeding this normally distributed dot product through another sine is also arcsine distributed for any $c > \sqrt{6}$. Note that the weights of a SIREN can be interpreted as angular frequencies while the biases are phase offsets. Thus, larger frequencies appear in the networks for weights with larger magnitudes. For $|\mathbf{w}^T\mathbf{x}| < \pi/4$, the sine layer will leave the frequencies unchanged, as the sine is approximately linear. In fact, we empirically find that a sine layer keeps spatial frequencies approximately constant for amplitudes such as $|\mathbf{w}^T\mathbf{x}| < \pi$, and increases spatial frequencies for amplitudes above this value[2].

Hence, we propose to draw weights with $c = \sqrt{6}$ so that $w_i \sim \mathcal{U}(-\sqrt{6/n}, \sqrt{6/n})$. This ensures that the input to each sine activation is normal distributed with a standard deviation of 1. Since only a few weights have a magnitude larger than $\pi$, the frequency throughout the sine network grows only slowly. Finally, we propose to initialize the first layer of the sine network with weights so that the sine function $\sin(\omega_0 \cdot \mathbf{W}\mathbf{x} + \mathbf{b})$ spans multiple periods over $[-1, 1]$. We found $\omega_0 = 30$ to work well for all the applications in this work. The proposed initialization scheme yielded fast and robust convergence using the ADAM optimizer for all experiments in this work.

## 4   Experiments

In this section, we leverage SIRENs to solve challenging boundary value problems using different types of supervision of the derivatives of $\Phi$. We first solve the Poisson equation via direct supervision of its derivatives. We then solve a particular form of the Eikonal equation, placing a unit-norm constraint on gradients, parameterizing the class of signed distance functions (SDFs). SIREN significantly

| ReLU (baseline) | SIREN (ours) | ReLU (baseline) | SIREN (ours) |

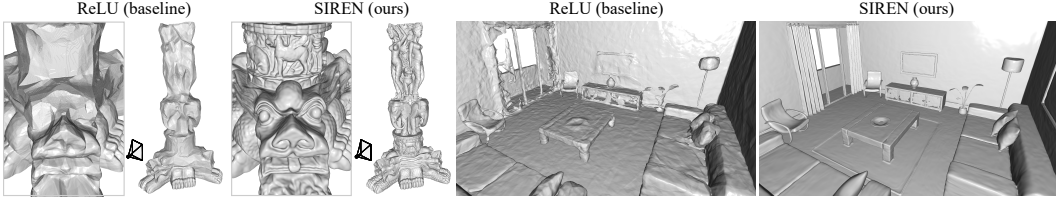

Figure 4: Shape representation. We fit signed distance functions parameterized by implicit neural representations directly on point clouds. Compared to ReLU implicit representations, our periodic activations significantly improve detail of objects (left) and complexity of entire scenes (right).

outperforms ReLU-based representations of SDFs, capturing large scenes at a high level of detail. We then solve the second-order Helmholtz partial differential equation, and the challenging inverse problem of full-waveform inversion. Finally, we combine SIRENs with hypernetworks, learning a prior over the space of parameterized functions. Those experiments are summarized in Section 4 of the supplemental, and additional experiments and details can be found in Section 5–11 in the supplemental. All code and data is publicly available on the project webpage[3].

## 4.1 Solving the Poisson Equation

We demonstrate that the proposed representation is not only able to accurately represent a function and its derivatives, but that it can also be supervised solely by its derivatives, i.e., the model is never presented with the actual function values, but only values of its first or higher-order derivatives.

An intuitive example representing this class of problems is the Poisson equation. The Poisson equation is perhaps the simplest elliptic partial differential equation (PDE) which is crucial in physics and engineering, for example to model potentials arising from distributions of charges or masses. In this problem, an unknown ground truth signal $f$ is estimated from discrete samples of either its gradients $\nabla f$ or Laplacian $\Delta f = \nabla \cdot \nabla f$ as

$$\mathcal{L}_{\text{grad.}} = \int_\Omega \|\nabla_\mathbf{x}\Phi(\mathbf{x}) - \nabla_\mathbf{x}f(\mathbf{x})\|\, \mathrm{d}\mathbf{x}, \quad \text{or} \quad \mathcal{L}_{\text{lapl.}} = \int_\Omega \|\Delta\Phi(\mathbf{x}) - \Delta f(\mathbf{x})\|\, \mathrm{d}\mathbf{x}. \quad (5)$$

**Poisson image reconstruction.** Solving the Poisson equation enables the reconstruction of images from their derivatives. We show results of this approach using SIREN in Fig. 3. Supervising the implicit representation with either ground truth gradients via $\mathcal{L}_{\text{grad.}}$ or Laplacians via $\mathcal{L}_{\text{lapl.}}$ successfully reconstructs the image. Remaining intensity variations are due to the ill-posedness of the problem.

**Poisson image editing.** Images can be seamlessly fused in the gradient domain [49]. For this purpose, $\Phi$ is supervised using $\mathcal{L}_{\text{grad.}}$ of Eq. (5), where $\nabla_\mathbf{x}f(\mathbf{x})$ is a composite function of the gradients of two images $f_{1,2}$: $\nabla_\mathbf{x}f(\mathbf{x}) = \alpha \cdot \nabla f_1(x) + (1-\alpha)\cdot\nabla f_2(x),\ \alpha \in [0,1]$. Fig. 3 shows two images seamlessly fused with this approach.

## 4.2 Representing Shapes with Signed Distance Functions

Inspired by recent work on shape representation with differentiable signed distance functions (SDFs) [1, 4, 9], we fit SDFs directly on oriented point clouds using both ReLU-based implicit neural representations and SIRENs. This amounts to solving a particular Eikonal boundary value problem that constrains the norm of spatial gradients $|\nabla_\mathbf{x}\Phi|$ to be 1 almost everywhere. Note that ReLU networks are seemingly ideal for representing SDFs, as their gradients are locally constant and their second derivatives are 0. Solving the Eikonal equation with an implicit neural representation with ReLU activations was previously proposed in [9]. We fit a SIREN to an oriented point cloud using a loss of the form

$$\mathcal{L}_{\text{sdf}} = \int_\Omega \big\| |\nabla_\mathbf{x}\Phi(\mathbf{x})| - 1 \big\|\, \mathrm{d}\mathbf{x} + \int_{\Omega_0} \|\Phi(\mathbf{x})\| + \big(1 - \langle\nabla_\mathbf{x}\Phi(\mathbf{x}), \mathbf{n}(\mathbf{x})\rangle\big)\, \mathrm{d}\mathbf{x} + \int_{\Omega\backslash\Omega_0} \psi\big(\Phi(\mathbf{x})\big)\, \mathrm{d}\mathbf{x}, \quad (6)$$

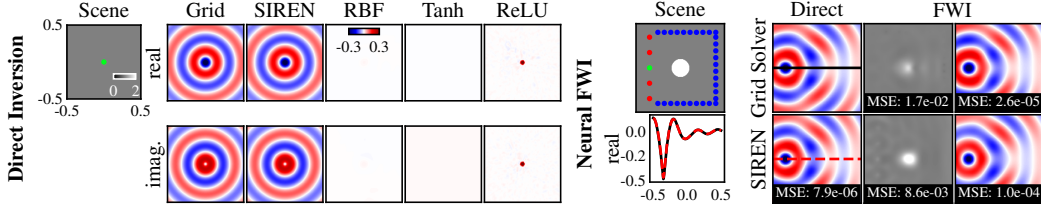

Figure 5: **Direct Inversion:** We solve the Helmholtz equation for a single point source placed at the center of a medium (green dot) with uniform wave propagation velocity (top left). The SIREN solution closely matches a principled grid solver [52] while other network architectures fail to find the correct solution. **Neural Full-Waveform Inversion (FWI):** A scene contains a source (green) and a circular wave velocity perturbation centered at the origin (top left). With the scene velocity known *a priori*, SIREN directly reconstructs a wavefield that closely matches a principled grid solver [52] (bottom left, middle left). For FWI, the velocity and wavefields are reconstructed with receiver measurements (blue dots) from sources triggered in sequence (green, red dots). The SIREN velocity model outperforms a principled FWI solver [53], accurately predicting wavefields. FWI MSE values are calculated across all wavefields and the visualized real wavefield corresponds to the green source.

Here, $\psi(\Phi(\mathbf{x})) = \exp(-\alpha \cdot |\Phi(\mathbf{x})|), \alpha \gg 1$ penalizes off-surface points for creating SDF values close to 0. $\Omega$ is the whole domain and we denote the zero-level set of the SDF as $\Omega_0$. The model $\Phi(\mathbf{x})$ is supervised using oriented points sampled on a mesh, where we require the SIREN to respect $\Phi(\mathbf{x}) = 0$ and its normals $\mathbf{n}(\mathbf{x}) = \nabla f(\mathbf{x})$. During training, each minibatch contains an equal number of points on and off the mesh, each one randomly sampled over $\Omega$. As seen in Fig. 4, the proposed periodic activations significantly increase the details of objects and the complexity of scenes that can be represented by these neural SDFs, parameterizing a full room from the ICL-NUIM dataset [50] with only a single five-layer fully connected neural network. This is in contrast to concurrent work that addresses the same failure of conventional MLP architectures to represent complex or large scenes by locally decoding a discrete representation, such as a voxelgrid, into an implicit neural representation [11, 12, 51]. We note that the resulting representations can be quite compact. For instance, the Thai statue shown in Figure 4 is reconstructed at a high fidelity while requiring only 260 kB while the naive mesh representation of this dataset requires 293 MB. Similarly, the SIREN representation of the room requires only about 1 MB whereas the naive mesh representation requires 579 MB. Please refer to the supplemental material for additional discussions on compression capabilities of SIREN.

### 4.3 Solving the Helmholtz and Wave Equations

The Helmholtz and wave equations are second-order partial differential equations related to the physical modeling of diffusion and waves. They are closely related through a Fourier-transform relationship, with the Helmholtz equation given as

$$H(m)\,\Phi(\mathbf{x}) = -f(\mathbf{x}), \text{ with } H(m) = \big(\Delta + m(\mathbf{x})\,w^2\big). \tag{7}$$

Here, $f(\mathbf{x})$ represents a known source function, $\Phi(\mathbf{x})$ is the unknown wavefield, and the squared slowness $m(\mathbf{x}) = 1/c(\mathbf{x})^2$ is a function of the wave velocity $c(\mathbf{x})$. In general, the solutions to the Helmholtz equation are complex-valued and require numerical solvers to compute. As the Helmholtz and wave equations follow a similar form, we discuss the Helmholtz equation here, with additional results and discussion for the wave equation in the supplement.

**Solving for the wavefield.** We solve for the wavefield by parameterizing $\Phi(\mathbf{x})$ with a SIREN. To accommodate a complex-valued solution, we configure the network to output two values, interpreted as the real and imaginary parts. Training is performed on randomly sampled points $\mathbf{x}$ within the domain $\Omega = \{\mathbf{x} \in \mathbb{R}^2 \,|\, \|\mathbf{x}\|_\infty < 1\}$. The network is supervised using a loss function based on the Helmholtz equation:

$$\mathcal{L}_{\text{Helmholtz}} = \int_\Omega \lambda(\mathbf{x})\,\|H(m)\Phi(\mathbf{x}) + f(\mathbf{x})\|\,\mathrm{d}\mathbf{x},$$

with $\lambda(\mathbf{x}) = k$, a hyperparameter, when $f(\mathbf{x}) \neq 0$ (corresponding to the inhomogeneous contribution to the Helmholtz equation) and $\lambda(\mathbf{x}) = 1$ otherwise (for the homogenous part). Each minibatch

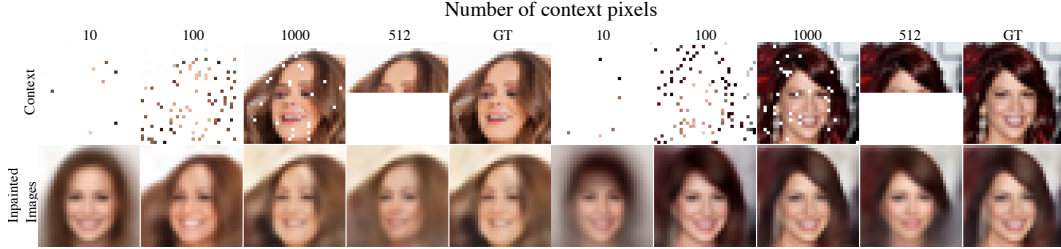

Figure 6: Generalizing across implicit functions parameterized by SIRENs on the CelebA dataset [56]. Image inpainting results are shown for various numbers of context pixels in $O_j$.

contains samples from both contributions and $k$ is set so the losses are approximately equal at the beginning of training. In practice, we use a slightly modified form of Equation (7) to include the perfectly matched boundary conditions that are necessary to ensure a unique solution [52] (see supplement for details).

Results are shown in Fig. 5 for solving the Helmholtz equation in two dimensions with spatially uniform wave velocity and a single point source (modeled as a Gaussian with $\sigma^2 = 10^{-4}$). The SIREN solution is compared with a principled solver [52] as well as other neural network solvers. All evaluated network architectures use the same number of hidden layers as SIREN but with different activation functions. In the case of the RBF network, we prepend a Gaussian RBF layer with 1024 hidden units and use a tanh activation for all the other layers. SIREN is the only representation capable of producing a high-fidelity reconstruction of the wavefield. We also note that the tanh network has a similar architecture to recent work on neural PDE solvers [44], except we increase the network size to match SIREN.

**Neural full-waveform inversion (FWI).**  In many wave-based sensing modalities (radar, sonar, seismic imaging, etc.), one attempts to probe and sense across an entire domain using sparsely placed sources (i.e., transmitters) and receivers. FWI uses the known locations of sources and receivers to jointly recover the entire wavefield and other physical properties, such as permittivity, density, or wave velocity. Specifically, the FWI problem can be described as [54]

$$\arg\min_{m,\Phi} \sum_{i=1}^{N} \int_{\Omega} \|\mathrm{III}_r(\Phi_i(\mathbf{x}) - r_i(\mathbf{x}))\|\, \mathrm{d}\mathbf{x} \text{ s.t. } H(m)\,\Phi_i(\mathbf{x}) = -f_i(\mathbf{x}),\ 1 \leq i \leq N,\ \forall \mathbf{x} \in \Omega, \quad (8)$$

where there are $N$ sources, $\mathrm{III}_r$ samples the wavefield at the receiver locations, and $r_i(x)$ models receiver data for the $i$th source.

We first use a SIREN to directly solve Eq. 7 for a known wave velocity perturbation, obtaining an accurate wavefield that closely matches that of a principled solver [52] (see Fig. 5, right). Without *a priori* knowledge of the velocity field, FWI is used to jointly recover the wavefields and velocity. Here, we use 5 sources and place 30 receivers around the domain, as shown in Fig. 5. Using the principled solver, we simulate the receiver measurements for the 5 wavefields (one for each source) at a single frequency of 3.2 Hz, which is chosen to be relatively low for improved convergence. We pre-train SIREN to output 5 complex wavefields and a squared slowness value for a uniform velocity. Then, we optimize for the wavefields and squared slowness using a penalty method variation [54] of Eq. 8 (see the supplement for additional details). In Fig. 5, we compare to an FWI solver based on the alternating direction method of multipliers [53, 55]. With only a single frequency for the inversion, the principled solver is prone to converge to a poor solution for the velocity. As shown in Fig. 5, SIREN converges to a better velocity solution and accurate solutions for the wavefields. All reconstructions are performed or shown at $256 \times 256$ resolution to avoid noticeable stair-stepping artifacts in the circular velocity perturbation.

## 4.4 Learning a Space of Implicit Functions

A powerful concept that has emerged for implicit representations is to learn priors over the space of functions that define them [1, 2, 10]. Here we demonstrate that the function space parameterized by SIRENs also admits the learning of powerful priors. Each of these SIRENs $\Phi_j$ are fully defined

by their parameters $\boldsymbol{\theta}_j \in \mathbb{R}^l$. Assuming that all parameters $\boldsymbol{\theta}_j$ of a class exist in a $k$-dimensional subspace of $\mathbb{R}^l$, $k < l$, then these parameters can be well modeled by latent code vectors in $\mathbf{z} \in \mathbb{R}^k$. Like in neural processes [57–59], we condition these latent code vectors on partial observations of the signal $O \in \mathbb{R}^m$ through an encoder

$$C : \mathbb{R}^m \to \mathbb{R}^k, \quad O_j \mapsto C(O_j) = \mathbf{z}_j, \tag{9}$$

and use a ReLU hypernetwork [60], to map the latent code to the weights of a SIREN, as in [10]:

$$\Psi : \mathbb{R}^k \to \mathbb{R}^l, \quad \mathbf{z}_j \mapsto \Psi(\mathbf{z}_j) = \boldsymbol{\theta_j}. \tag{10}$$

We replicated the experiment from [57] on the CelebA dataset [56] using a set encoder. Additionally, we show results using a convolutional neural network encoder which operates on sparse images. Interestingly, this improves the quantitative and qualitative performance on the inpainting task. At test time, this enables reconstruction from sparse pixel observations, and, thereby, inpainting. Fig. 6 shows test-time reconstructions from a varying number of pixel observations. Note that these inpainting results were all generated using the same model, with the same parameter values. Tab. 1 reports a quantitative comparison to [57], demonstrating that generalization over SIREN representations is at least equally as powerful as generalization over images.

Table 1: Quantitative comparison to Conditional Neural Processes [57] (CNPs) on the $32 \times 32$ CelebA test set. The pixel-wise mean squared errors are reported.

| Number of Context Pixels | 10 | 100 | 1000 |
|---|---|---|---|
| CNP [57] | 0.039 | 0.016 | 0.009 |
| Set Encoder + Hypernet. | 0.035 | 0.013 | 0.009 |
| CNN Encoder + Hypernet. | **0.033** | **0.009** | **0.008** |

## 5   Discussion and Conclusion

The question of how to represent a signal is at the core of many problems across science and engineering. Implicit neural representations may provide a new tool for many of these by offering a number of potential benefits over conventional continuous and discrete representations. We demonstrate that periodic activation functions are ideally suited for representing complex natural signals and their derivatives using implicit neural representations. We also prototype several boundary value problems that our framework is capable of solving robustly. There are several exciting avenues for future work, including the exploration of other types of inverse problems and applications in areas beyond implicit neural representations, for example neural ODEs [46]. While we demonstrate the feasibility of generalizing across signals represented by SIREN networks, the fidelity of the resulting representations is limited—investigating effective alternatives is an important direction for future work. An immediate application of SIREN may be the compression of large-scale 3D models, as SIREN may represent them at a high visual fidelity with a relativily small number of parameters and resulting small file sizes.

Concurrent work investigates directions related to our approach. Locally decoding a discrete voxelgrid into an implicit neural representation [11, 12, 51] similarly enables the representation of fine detail. Tancik et al. [61] extend the previously proposed first-layer positional encoding [5, 48] and investigate its properties through the perspective of the neural tangent kernel [62].

## Broader Impact

The proposed SIREN representation enables accurate representations of natural signals, such as images, audio, and video in a deep learning framework. This may be an enabler for downstream tasks involving such signals, such as classification for images or speech-to-text systems for audio. Such applications may be leveraged for both positive and negative ends. SIREN may in the future further enable novel approaches to the generation of such signals. This has potential for misuse in impersonating actors without their consent. For an in-depth discussion of such so-called DeepFakes, we refer the reader to a recent review article on neural rendering [16].

## Acknowledgments and Disclosure of Funding

V.S., A.W.B., and D.B.L. were supported by a Stanford Graduate Fellowship. J.N.P.M was supported by a Swiss National Science Foundation Fellowship (P2EZP2-181817). G.W. was supported by a Sloan Fellowship, by the NSF (award numbers 1553333 and 1839974), and a PECASE by the ARO.

## Footnotes

[2]Formalizing the distribution of output frequencies throughout SIRENs proves to be a hard task and is out of the scope of this work.

[3]https://vsitzmann.github.io/siren/

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
