[Supplementary Material · Siren_supplement.pdf]

# Implicit Neural Representations with Periodic Activation Functions

# –Supplementary Material–

**Vincent Sitzmann**[*]
sitzmann@cs.stanford.edu

**Julien N. P. Martel**[*]
jnmartel@stanford.edu

**Alexander W. Bergman**
awb@stanford.edu

**David B. Lindell**
lindell@stanford.edu

**Gordon Wetzstein**
gordon.wetzstein@stanford.edu

Stanford University
vsitzmann.github.io/siren/

## Contents

---

[*]These authors contributed equally to this work.

# 1 Initialization and Distribution of Activations

## 1.1 Informal statement

Initialization schemes have been shown to be crucial in the training procedure of deep neural networks [20, 18]. Here, we propose an initialization scheme for SIREN that preserves the distribution of activations through its layers and thus allows us to build deep architectures.

**Statement of the initialization scheme.** We propose to draw weights according to a uniform distribution $W \sim \mathcal{U}(-\sqrt{6/\text{fan\_in}}, \sqrt{6/\text{fan\_in}})$. We claim that this leads to the input of each sine activation being Gauss-Normal distributed, and the output of each sine activation approximately arcsine-distributed with a standard deviation of $0.5$. Further, we claim that the form as well as the moments of these distributions do not change as the depth of the network grows.

**Overview of the proof.** Our initialization scheme relies on the fact that if the input to a neuron in a layer is distributed the same way as its output, then by a simple recursive argument we can see that the distributions will be preserved throughout the network.

Hence, we consider an input in the interval $[-1, 1]$. We assume it is drawn uniformly at random, since we interpret it as a "normalized coordinate" in our applications. We first show in **Lemma 1.1**, that pushing this input through a sine nonlinearity yields an arcsine distribution. The second layer (and, as we will show, all following layers), computes a linear combination of such arcsine distributed outputs (of known variance, **Lemma 1.3**). Following Lindeberg's condition for the central limit theorem, this linear combination will be normal distributed **Lemma 1.5**, with a variance that can be calculated using the variance of the product of random variables (**Lemma 1.4**). It remains to show that pushing a Gaussian distribution through the sine nonlinearity again yields an arcsine distributed output **Lemma 1.6**, and thereby, we may apply the same argument to the distributions of activations of the following layers.

We formally present the lemmas and their proof in the next section before formally stating the initialization scheme and proving it in Section 1.3. We show empirically that the theory predicts very well the behaviour of the initialization scheme in Section 1.4.

## 1.2 Preliminary results

First let us note that the sine function is periodic, of period $2\pi$ and odd: $\sin(-x) = -\sin(x)$, i.e. it is symmetric with respect to the origin. Since we are interested in mapping "coordinates" through SIREN, we will consider an input as a random variable $X$ uniformly distributed in [-1,1]. We will thus study, without loss of generality, the frequency scaled SIREN that uses the activation $\sin\left(\frac{\pi}{2}x\right)$. Which is half a period (note that the distribution does not change on a full period, it is "just" considering twice the half period).

**Definition 1.1.** *The arcsine distribution is defined for a random variable $X$ by its cumulative distribution function (CDF) $F_X$ such as*

$$X \sim \text{Arcsin}(a, b), \text{ with CDF: } F_X(x) = \frac{2}{\pi} \arcsin\left(\sqrt{\frac{x-a}{b-a}}\right), \text{with } b > a.$$

**Lemma 1.1.** *Given $X \sim \mathcal{U}(-1, 1)$, and $Y = \sin\left(\frac{\pi}{2}X\right)$ we have $Y \sim \text{Arcsin}(-1, 1)$.*

*Proof.* The cumulative distribution function (CDF) $F_X(x) = \mathbb{P}(X \leq x)$ is defined, for a random variable that admits a continuous probability density function (PDF), $f$ as the integral $F_X(x) = \int_{-\infty}^{x} f(t) \, dt$. Hence, for the uniform distribution $\mathcal{U}(-1, 1)$ which is $f(x) = \frac{1}{2}$ over the interval $[-1, 1]$ and $0$ everywhere else, it is easy to show that: $F_X(x) = \frac{1}{2}x + \frac{1}{2}$.

We are interested in the distribution of the output $Y = \sin\left(\frac{\pi}{2}X\right)$. Noting that $\sin\left(\frac{\pi}{2}\right)$ is bijective on $[-1, 1]$, we have

$$F_Y(y) = \mathbb{P}(\sin\left(\frac{\pi}{2}X\right) \leq y) = \mathbb{P}(X \leq \frac{2}{\pi} \arcsin y) = F_X\left(\frac{2}{\pi} \arcsin y\right),$$

Substituting the CDF $F_X$, noting it is the uniform distribution which has a compact support (this is [-1,1]), we have

$$F_Y(y) = \frac{1}{\pi} \arcsin y + \frac{1}{2}.$$

Using the identity $\arcsin \sqrt{x} = \frac{1}{2} \arcsin(2x - 1) + \frac{\pi}{4}$, we conclude:

$$F_Y(y) \sim \text{Arcsin}(-1, 1).$$

The PDF can be found, deriving the cdf: $f_Y(y) = \frac{d}{dy} F_Y(y) = \frac{1}{\pi} \frac{1}{\sqrt{1-y^2}}.$ $\qquad\square$

**Lemma 1.2.** *The variance of $mX + n$ with X a random variable and $m \in \mathbb{R}^+_{/0}, n \in \mathbb{R}$ is $\text{Var}[mX + n] = m^2\text{Var}[X]$.*

*Proof.* For any random variable with a continuous pdf $f_X$, its expectation is defined as $\text{E}[X] = \int_{-\infty}^{\infty} f_X(x)dx$. The variance is defined as $\text{Var}[X] = \text{E}[(X - \text{E}[X])^2] = \text{E}[X^2] - \text{E}[X]^2$. Thus, we have $\text{Var}[mX + n] = \text{E}[(mX + n)^2] - \text{E}[mX + n]^2 = \text{E}[m^2X^2 + 2mnX + n^2] - (m\text{E}[X] + n)^2 = m^2(\text{E}[X^2] - \text{E}[X]^2) = m^2\text{Var}[X]$. $\qquad\square$

**Lemma 1.3.** *The variance of $X \sim \text{Arcsin}(a, b)$ is $\text{Var}[X] = \frac{1}{8}(b - a)^2$.*

*Proof.* First we prove that if $Z \sim \text{Arcsin}(0, 1)$ then $\text{Var}[Z] = \frac{1}{8}$. We have $\text{E}[Z] = \frac{1}{2}$ by symmetry, and $\text{Var}[Z] = \text{E}[Z^2] - \text{E}[Z]^2 = \text{E}[Z^2] - \frac{1}{4}$. Remains to compute:

$$\text{E}[Z^2] = \int_0^1 z^2 \cdot \frac{1}{\pi\sqrt{z(1-z)}} \, dz = \frac{2}{\pi} \int_0^1 \frac{t^4}{\sqrt{1-t^2}} \, dt = \frac{2}{\pi} \int_0^{\pi/2} \sin^4 u \, du = \frac{3}{8},$$

using a first change of variable: $z = t^2$, $dz = 2t \, dt$ and then a second change of variable $t = \sin(u), dt = \cos(u)du$. The integral of $\sin^4(u)$ is calculated remarking it is $(sin^2(u))^2$, and using the formulas of the double angle: $\cos(2u) = 2\cos^2(u) - 1 = 1 - 2\sin^2(u)$.

Second, we prove that if $X \sim \text{Arcsin}(\alpha, \beta)$ then the linear combination $mX + n \sim \text{Arcsin}(\alpha m + n, \beta m + n)$ with $m \in \mathbb{R}_{/0}, n \in \mathbb{R}$, (using the same method as in **Lemma 1.1** with $Y = mX + n$).

Posing $X = mZ + n$ and using $n = a$ and $m = b - a$, we have $X \sim \text{Arcsin}(m \cdot 0 + n, m \cdot 1 + n) = \text{Arcsin}(a, b)$. Finally, $\text{Var}[X] = \text{Var}[m \cdot Z + n] = m^2 \cdot \text{Var}[Z] = (b - a)^2 \cdot \frac{1}{8}$ (**Lemma 1.2**). $\qquad\square$

**Lemma 1.4.** *For two independent random variables X and Y*

$$\text{Var}[X \cdot Y] = \text{Var}[X] \cdot \text{Var}[Y] + \text{E}[Y]^2 \cdot \text{Var}[X] + \text{E}[X]^2 \cdot \text{Var}[Y].$$

*Proof.* See [19]. $\qquad\square$

**Theorem 1.5.** *Central Limit Theorem with Lindeberg's sufficient condition. Let $X_k$, $k \in \mathbb{N}$ be independent random variables with expected values $\text{E}[X_k] = \mu_k$ and variances $\text{Var}[X_k] = \sigma_k$. Posing $s_n^2 = \sum_{k=1}^{n} \sigma_k^2$. If the $X_k$ statisfy the Lindenberg condition:*

$$\lim_{n\to\infty} \frac{1}{s_n^2} \sum_{k=1}^{n} \text{E}[(X_k - \mu_k)^2 \cdot \mathbf{1}(|X_k - \mu_k| > \epsilon s_n)] = 0 \qquad (1)$$

*$\forall \epsilon > 0$, then the Central Limit Theorem (CLT) holds. That is,*

$$S_n = \frac{1}{s_n} \sum_{k=1}^{n} (X_k - \mu_k), \qquad (2)$$

*converges in distribution to the standard normal distribution as $n \to \infty$.*

*Proof.* See [27, 2]. $\qquad\square$

**Lemma 1.6.** *Given a Gaussian distributed random variable $X \sim \mathcal{N}(0, 1)$ and $Y = \sin \frac{\pi}{2} X$ we have $Y \sim \text{Arcsin}(-1, 1)$.*

Figure 1: **Top left:** A plot of the standard normal distribution on $[-3, 3]$ as well as the graph of $y = \sin\frac{\pi}{2}x$ and its three reciprocal bijections $y = \frac{2}{\pi}\arcsin(-x-2)$, $y = \frac{2}{\pi}\arcsin x$ and $y = \frac{2}{\pi}\arcsin(2-x)$ covering the interval $[-3, 3]$ in which $99.7\%$ of the probability mass of the standard normal distribution lies. **Bottom left:** Plot of the approximation of the CDF of the standard normal with a logistic function. **Right:** Comparison of the theoretically derived CDF of the output of a sine nonlinearity (green) and the ground-truth Arcsine CDF (red), demonstrating that a standard normal distributed input fed to a sine indeed yields an approximately Arcsine distributed output.

*Proof.* For a random variable $X$ normally distributed we can approximate the CDF of its normal distribution with the logistic function, as in [10]:

$$F_X(x) = \frac{1}{2} + \frac{1}{2}\text{erf}(\frac{x}{\sqrt{2}})$$

$$\approx \left(1 + \exp(-\alpha \cdot x)\right)^{-1}$$

$$\approx \frac{1}{2} + \frac{1}{2}\tanh(\beta \cdot x),$$

with $\alpha = 1.702$ and $\beta = 0.690$. Similar to the proof of Lemma (1.1), we are looking for the CDF of the random variable $Y \sim \sin\left(\frac{\pi}{2}X\right)$. However, the normal distribution does not have compact support. This infinite support yields an infinite series describing the CDF of Y.

Hence, we make a second approximation that consists in approximating the CDF of Y on the interval $[-3, 3]$. Because $X \sim \mathcal{N}(0, 1)$, we know that $99.7\%$ of the probability mass of $X$ lies on the compact set $[-3, 3]$. Thus, ignoring the other contributions, we have:

$$F_Y(y) = \mathbb{P}(\sin\left(\frac{\pi}{2}X\right) \leq y)$$

$$= F_X(3) - F_X\left(2 - \frac{2}{\pi}\arcsin x\right) + F_X\left(\frac{2}{\pi}\arcsin x\right) - F_X\left(-\frac{2}{\pi}\arcsin x - 2\right).$$

Using the logistic approximation of the CDF of $X$, this is:

$$F_X(x) = \frac{1}{2}\tanh(3\beta)$$

$$+ \frac{1}{2}\left(\tanh\left(\frac{2\beta}{\pi}z\right) - \tanh\left(2\beta(1 - \frac{1}{\pi}z)\right) - \tanh\left(-2\beta(1 + \frac{1}{\pi}z)\right)\right),$$

with $z = \arcsin x$. Using a taylor expansion in $z = 0$ (and noting that $\arcsin 0 = 0$) we have:

$$F_X(x) \stackrel{0}{=} \frac{1}{2}\tanh(3\beta) + \frac{1}{\pi} \cdot \arcsin x,$$

which approximates $X \sim \text{Arcsin}(-1, 1)$. Figure 1 illustrates the different steps of the proofs and the approximations we made. □

**Lemma 1.7.** *The variance of $X \sim \mathcal{U}(-a, b)$ is* $\text{Var}[X] = \frac{1}{12}(b-a)^2$

*Proof.* $\text{E}[X] = \frac{a+b}{2}$. $\text{Var}[X] = \text{E}[X^2] - \text{E}[X]^2 = \frac{1}{b-a}[\frac{x^3}{3}]_a^b - (\frac{a+b}{2})^2 = \frac{1}{b-a}\frac{b^3-a^3}{3} - (\frac{a+b}{2})^2$, developing the cube as $b^3 - a^3 = (b-a)(a^2 + ab + b^2)$ and simplifying yields the result. □

### 1.3 Formal statement and proof of the initialization scheme

**Theorem 1.8.** *For a uniform input in* $[-1, 1]$*, the activations throughout a* SIREN *are standard normal distributed before each sine nonlinearity and arcsine-distributed after each sine nonlinearity, irrespective of the depth of the network, if the weights are distributed uniformly in the interval* $[-c, c]$ *with* $c = \sqrt{6/fan\_in}$ *in each layer.*

*Proof.* Assembling all the lemma, a sketch of the proof is:

- Each output $X_l$ for the layer $l$ is $X_l \sim \text{Arcsin}(-1, 1)$ (first layer: from a uniform distribution **Lemma 1.1**, next layers: from a standard-normal **Lemma 1.6**) and $\text{Var}[X_l] = \frac{1}{2}$ (**Lemma 1.3**).

- The input to the layer $l + 1$ is $w_l^T X_l = \sum_i^n w_{i,l} X_{i,l}$ (bias does not change distribution for high enough frequency). Using weights $w_i^l \sim \mathcal{U}(-c, c)$ we have $\text{Var}[w_l^T X_l] = \text{Var}[w_l] \cdot \text{Var}[X_l] = \frac{1}{12}(2c)^2 \cdot \frac{1}{2} = \frac{1}{6}c^2$ (from the variance of a uniform distribution **Lemma 1.7**, and an arcsine distribution **Lemma 1.3**, as well as their product **Lemma 1.4**).

- Choosing $c = \sqrt{\frac{6}{n}}$, with the fan-in $n$ (see dot product above) and using the CLT with weak Lindenberg's condition we have $\text{Var}[w_l^T X_l] = n \cdot \frac{1}{6} \frac{6}{n} = 1$ **Lemma 1.5** and $w_l^T X_l \sim \mathcal{N}(0, 1)$

- This holds true for all layers, since normal distribution through the sine non-linearity yields again the arcsine distribution **Lemma 1.2**, **Lemma 1.6**

$\square$

### 1.4 Empirical evaluation

We validate our theoretical derivation with an experiment. We assemble a 6-layer, single-input SIREN with 2048 hidden units, and initialize it according to the proposed initialization scheme. We draw $2^8$ inputs in a linear range from $-1$ to $1$ and plot the histogram of activations after each linear layer and after each sine activation. We further compute the 1D Fast Fourier Transform of all activations in a layer. Lastly, we compute the sum of activations in the final layer and compute the gradient of this sum w.r.t. each activation. The results can be visualized in Figure 2. The distribution of activations nearly perfectly matches the predicted Gauss-Normal distribution after each linear layer and the arcsine distribution after each sine nonlinearity. As discussed in the main text, frequency components of the spectrum similarly remain comparable, with the maximum frequency growing only slowly. We verified this initialization scheme empirically for a 50-layer SIREN with similar results. Finally, similar to the distribution of activations, we plot the distribution of gradients and empirically demonstrate that it stays almost perfectly constant across layers, demonstrating that SIREN does not suffer from either vanishing or exploding gradients at initialization. We leave a formal investigation of the distribution of gradients to future work.

## 2 Remarks about Training

All our networks were trained using ADAM [23]. In the following we compile experiments and remarks about training SIREN.

### 2.1 Network depth

We performed additional experiments in which we increased the depth of our network from 3 hidden layers (in the main paper) to 5, 10 and 15 layers of 256 units below, and plotted the accuracy of the resulting network when solving the Poisson equation on the Cameraman image. Increasing the network depth yields a higher accuracy (measured as a L2 loss) as well as a much faster convergence rate as can be seen in Figure 3

Figure 2: Activation and gradient statistics at initialization for a 6-layer SIREN. Increasing layers from top to bottom. Orange dotted line visualizes analytically predicted distributions. Note how the experiment closely matches theory, activation distributions stay consistent from layer to layer, the maximum frequency throughout layers grows only slowly, and gradient statistics similarly stay consistent from layer to layer.

Figure 3: Accuracy (measured as mean L2 loss) vs. network depth, fitting 5, 10, 15 layers (256 units each) SIRENs solving the Poisson Equation (i.e. trained via the gradient of the network on the gradient of the image) on the MIT Cameraman image.

## 2.2 Spikes in the learning curves

In Figure 3, one can observe spikes in the learning rate (green curve for 15 layers). A peculiarity of SIRENs is that despite shortly getting worse, the networks seems to consistently recover from these error spikes and continue increasing their accuracy. We hypothesize the recovery behaviour comes from the bounded and periodic nature of the sine non-linearity. We have observed the intensity and frequency of those error spikes can be reduced by tuning down the learning rate.

## 2.3 Normalization layers

We experimented with a variety of normalization layers. We note that the sine activation has normalizing properties, and the proposed initialization scheme guarantees standard normal distributed activations at initialization (see suppl.). Likely as a result of this, normalization layers did not lead to performance gains.

## 3   Evaluating the Gradient of a SIREN is Evaluating another SIREN

We can write a loss $L$ between a target and a SIREN output as:

$$L\big(\text{target}, (\mathbf{W}_n \circ \phi_{n-1} \circ \phi_{n-2} \ldots \phi_0)(\mathbf{x}) + \mathbf{b}_n\big). \tag{3}$$

A sine layer is defined as:

$$\phi_i(\mathbf{x}) = (\sin \circ \mathbf{T}_i)(\mathbf{x}), \quad \text{with } \mathbf{T}_i : \mathbf{x} \mapsto \mathbf{W}_i \mathbf{x} + \mathbf{b}_i = \hat{\mathbf{W}}_i \hat{\mathbf{x}}, \tag{4}$$

defining $\hat{\mathbf{W}} = [\mathbf{W}, \mathbf{b}]$ and $\hat{\mathbf{x}} = [\mathbf{x}, 1]$ for convenience.

The gradient of the loss with respect to the input can be calculated using the chain rule:

$$\nabla_{\mathbf{x}} L = \left( \frac{\partial L}{\partial \mathbf{y}_n} \cdot \frac{\partial \mathbf{y}_n}{\partial \mathbf{y}_{n-1}} \cdot \ldots \frac{\partial \mathbf{y}_1}{\partial \mathbf{y}_0} \cdot \frac{\partial \mathbf{y}_0}{\partial \mathbf{x}} \right)^T$$

$$= (\hat{\mathbf{W}}_0^T \cdot \sin'(\mathbf{y}_0)) \cdot \ldots \cdot (\hat{\mathbf{W}}_{n-1}^T \cdot \sin'(\mathbf{y}_{n-1})) \cdot \hat{\mathbf{W}}_n^T \cdot L'(\mathbf{y}_n) \tag{5}$$

where $\mathbf{y}_l(\hat{\mathbf{x}})$ is defined as the network evaluated on input $\hat{\mathbf{x}}$ stopping before the non-linearity of layer $l$, ($\hat{\mathbf{x}}$ is implicit in Equation (5) for the sake of readability):

$$\mathbf{y}_0(\hat{\mathbf{x}}) = \hat{\mathbf{W}}_0 \hat{\mathbf{x}}$$

$$\mathbf{y}_l(\hat{\mathbf{x}}) = (\hat{\mathbf{W}}_l \circ \sin)(\mathbf{y}_{l-1}) = (\hat{\mathbf{W}}_l \circ \sin \circ \ldots \hat{\mathbf{W}}_0)(\hat{\mathbf{x}}) \tag{6}$$

Remarking that the derivative $\sin'(\mathbf{y}_l) = \cos(\mathbf{y}_l) = \sin(\mathbf{y}_l + \frac{\pi}{2})$, and that we can absorb the $\frac{\pi}{2}$ phase offset in the bias by defining the new weight matrix $\check{\mathbf{W}} = [\mathbf{W}, \mathbf{b} + \frac{\pi}{2}]$. The gradient can be rewritten:

$$\nabla_{\mathbf{x}} L = (\hat{\mathbf{W}}_0^T \cdot \sin(\check{\mathbf{y}}_0)) \cdot \ldots \cdot (\hat{\mathbf{W}}_{n-1}^T \cdot \sin(\check{\mathbf{y}}_{n-1})) \cdot \hat{\mathbf{W}}_n^T \cdot L'(\mathbf{y}_n) \tag{7}$$

**Figure 4:** Poisson image reconstruction using the ReLU P.E. (left) and tanh (right) network architectures. For both architectures, image reconstruction from the gradient is of lower quality than SIREN, while reconstruction from the Laplacian is not at all accurate.

with $\check{\mathbf{y}}_l$ the activations using the weights $\check{\mathbf{W}}_l$

$$\check{\mathbf{y}}_0(\hat{\mathbf{x}}) = \check{\mathbf{W}}_0\hat{\mathbf{x}}$$
$$\check{\mathbf{y}}_l(\hat{\mathbf{x}}) = (\check{\mathbf{W}}_l \circ \sin)(\mathbf{y}_{l-1}) = (\check{\mathbf{W}}_l \circ \sin \circ \ldots \check{\mathbf{W}}_0)(\hat{\mathbf{x}}) \tag{8}$$

which is a forward pass evaluating a slightly different SIREN in which all the biases have been shifted by $\frac{\pi}{2}$.

Furthermore, in Equation (7) since every term of the form $\sin(\mathbf{y}_l)$ is a SIREN, and those terms are multiplied by weight matrices between them, this shows that the gradient of a SIREN can be evaluated by yet another SIREN. It also shows that a SIREN of L non linearities, requires the evaluation of a SIREN-like network (whose computation graph is not a simple chain but only contains sine non-linearities) of $\sum_{l=1}^{N} l = \frac{L \cdot (L+1)}{2}$ non-linearities.

## 4 Experiment summary

**Table 1:** Overview of the input, outputs, losses and boundary values used in the experiments performed in our paper.

| Name | Input | Output | Loss | Boundary values |
|---|---|---|---|---|
| Image | $\mathbf{x} \in \mathbb{R}^2$ | $\mathbb{R}$ | $\int_\Omega \|\Phi(\mathbf{x}) - f(\mathbf{x})\| \, \mathrm{d}\mathbf{x}$ | n/a |
| Audio | $t \in \mathbb{R}$ | $\mathbb{R}$ | $\int_\Omega \|\Phi(t) - f(t)\| \, \mathrm{d}t$ | n/a |
| Video | $(\mathbf{x}, t) \in \mathbb{R}^3$ | $\mathbb{R}$ | $\int_\Omega \|\Phi(\mathbf{x}, t) - f(\mathbf{x}, t)\| \, \mathrm{d}\mathbf{x} \, \mathrm{d}t$ | n/a |
| Poisson grad. | $\mathbf{x} \in \mathbb{R}^2$ | $\mathbb{R}$ | $\int_\Omega \|\nabla\Phi(\mathbf{x}) - \nabla f(\mathbf{x})\| \, \mathrm{d}\mathbf{x}$ | none |
| Poisson lapl. | $\mathbf{x} \in \mathbb{R}^2$ | $\mathbb{R}$ | $\int_\Omega \|\Delta\Phi(\mathbf{x}) - \Delta f(\mathbf{x})\| \, \mathrm{d}\mathbf{x}$ | none |
| Eikonal eqn. | $\mathbf{x} \in \mathbb{R}^3$ | $\mathbb{R}$ | see Eqn. (**??**) | $\Phi(\mathbf{x}) = 0, \mathbf{x} \in \Omega_0$ |
| Helmholtz eqn. | $\mathbf{x} \in \mathbb{R}^2$ | $\mathbb{C}$ | $\int_\Omega \|(\Delta + m(\mathbf{x})\omega^2)\Phi(\mathbf{x}) - f(\mathbf{x})\| \, \mathrm{d}\mathbf{x}$ | PML (suppl. Sec. 5.1) |
| Wave eqn. | $(\mathbf{x}, t) \in \mathbb{R}^2$ | $\mathbb{C}$ | $\int_\Omega \|\partial_t^2\Phi(\mathbf{x}, t) - c^2 \Delta\Phi(\mathbf{x}, t)\| \, \mathrm{d}\mathbf{x} \, \mathrm{d}t$ | $\partial\Phi_t(\mathbf{x}, 0) = 0$, and $\Phi(\mathbf{x}, t) = f(\mathbf{x})$ |

## 5 Solving the Poisson Equation

### 5.1 Architecture Comparisons

To show that our representation is unique in being able to represent signals while being supervised solely by their derivatives, we test other neural network architectures and activation functions on the Poisson image reconstruction task. We show that the performance of the ReLU P.E. network architecture, which performed best on the single image fitting task besides SIREN, is not as accurate in supervising on derivatives. This is shown in Fig. 4. Additionally, in Tab. 2, we compare the PSNR of the reconstructed image, gradient image, and Laplace image between various architectures for Poisson image reconstruction.

Table 2: Quantitative comparison of reconstructed image, gradient image, and Laplace image in the Poisson image reconstruction task on the starfish image. Reconstruction accuracy is reported in PSNR after the images have been colorized and normalized.

| Model | Tanh | | ReLU P.E. | | SIREN | |
|---|---|---|---|---|---|---|
| Supervised on | Grad. | Laplacian | Grad. | Laplacian | Grad. | Laplacian |
| Reconstructed Image | 25.79 | 7.11 | 26.35 | 11.14 | **32.91** | **14.95** |
| Reconstructed Grad. | 19.11 | 11.14 | 19.33 | 11.35 | **46.85** | **23.45** |
| Reconstructed Laplacian | 18.59 | 16.35 | 14.24 | 18.31 | **19.88** | **57.13** |

One interesting observation from Tab. 2 is that other architectures such as ReLU P.E. have trouble fitting the Laplace and gradient images even when directly supervised on them, despite being able to fit images relatively accurately. This may be because the ground truth gradient and Laplace images have many high frequency features which are challenging to represent with any architecture besides SIRENs. In the normalized and colorized images (which PSNR is computed upon), the gradient image fit with ReLU P.E. has "grainy" effects which are more noticeable on gradient and Laplacian images than on natural images.

## 5.2 Implementation & Reproducibility Details

**Data.** We use the BSDS500 [34], which we center-crop to $321 \times 321$ and resize to $256 \times 256$. The starfish image is the 19th image from this dataset. We will make the bear and pyramid images used in the Poisson image editing experiment publicly available with our code. The ground truth gradient image is computed using the Sobel filter, and is scaled by a constant factor of 10 for training. The ground truth Laplace image is computed using a Laplace filter, and is scaled by a constant factor of 10,000 for training.

**Architecture.** We use the same 5-layer SIREN MLP for all experiments on fitting images and gradients.

**Hyperparameters.** We train for 10,000 iterations, and at each iteration fit on every pixel in the gradient or Laplacian image. We use the Adam optimizer with a learning rate of $1 \times 10^{-4}$ for all experiments, including the Poisson image editing experiments.

**Runtime.** We train for 10,000 iterations, requiring approximately 90 minutes to fit and evaluate a SIREN.

**Hardware.** The networks are trained using NVIDIA Quadro RTX 6000 GPUs with 24 GB of memory.

## 6 Representing Shapes with Signed Distance Functions

We performed an additional baseline using the ReLU positional encoding [36] shown in Figure 5. Similar to the results we obtained using the ReLU positional encoding on images, zero-level set of the SDF, in which the shape is encoded features high-frequencies that are not present while the level of details remains low (despite being much higher that in ReLU, see main paper).

**Data.** We use the Thai statue from the The Stanford 3D Scanning Repository (http://graphics.stanford.edu/data/3Dscanrep/). The room is a Free 3D model from Turbosquid.com. We sample each mesh by subdividing it until we obtain 10 million points and their normals. Those are then converted in .xyz formats we load from our code.

**Evaluation.** To evaluate the Chamfer distance, we sample 30 thousand points from our represented meshes, as in [37]. Although changing the number of sampled points can affect the Chamfer values, the same trend is observed.

<div style="text-align: center;">

SIREN (ours)　　　　　ReLU PE (baseline)

</div>

Figure 5: A comparison of SIREN used to fit a SDF from an oriented point cloud against the same fitting performed by an MLP using a ReLU PE (proposed in [36]). Chamfer distance computed on a [-1,1] box for the Thai statue: ReLU=3.78e-4 (see paper), ReLU PE=6.19e-5, SIREN=5.50e-5. For the room representation shown in the paper, we observe the same trend in Chamfer distance computed on a [-1,1] box: ReLU=4.33e-4, SIREN=3.99e-4.

**Architecture.** We use the same 5-layer SIREN MLP for all experiments on SDF, using 256 units in each layer for the statue and 1024 units in each layer for the room.

**Hyperparameters.** We train for 50,000 iterations, and at each iteration fit on every voxel of the volume. We use the Adam optimizer with a learning rate of $1 \times 10^{-4}$ for all experiments. We use the cost described in our paper:

$$\mathcal{L}_{\text{sdf}} = \lambda_1 \int_\Omega \big\| \, |\nabla_{\mathbf{x}} \Phi(\mathbf{x})| - 1 \big\| d\mathbf{x} + \int_{\Omega_0} \lambda_2 \, \|\Phi(\mathbf{x})\| + \lambda_3 \big( 1 - \langle \nabla_{\mathbf{x}} \Phi(\mathbf{x}), \mathbf{n}(\mathbf{x}) \rangle \big) d\mathbf{x} + \lambda_2 \int_{\Omega \setminus \Omega_0} \psi\big(\Phi(\mathbf{x})\big) d\mathbf{x},$$

(9)

with the Eikonal constraint (gradient = 1) multiplied by $\lambda_1 = 5 \cdot 10^1$, the signed distance function constraint as well as the off-surface penalization (the regularizer) multiplied by $\lambda_2 = 3 \cdot 10^3$, and the oriented surface/normal constraint multiplied by $\lambda_3 = 1 \cdot 10^2$.

**Runtime.** We train for 50,000 iterations, requiring approximately 6h hours to fit and evaluate a SIREN. Though, we remark that SIREN converge already very well after around 5,000-7,000 iterations, much more iterations are needed for the baselines, hence the number of 50,000 iterations.

**Hardware.** The networks are trained using NVIDIA GTX Titan X GPUs with 12 GB of memory.

## 7 Solving the Helmholtz and Wave Equations

The Helmholtz and wave equations are second-order partial differential equations related to the physical modeling of diffusion and waves. They are closely related through a Fourier-transform relationship, with the Helmholtz equation given as

$$\underbrace{\left(\nabla^2 + m(\mathbf{x})w^2\right)}_{H(m)} \Phi(\mathbf{x}) = -f(\mathbf{x}).$$

(10)

Here, $f(\mathbf{x})$ represents a known source function, $\Phi(\mathbf{x})$ is the unknown wavefield, and the squared slowness $m(\mathbf{x}) = 1/c(\mathbf{x})^2$ is a function of the wave velocity $c(\mathbf{x})$. In general, the solutions to the Helmholtz equation are complex-valued and require numerical solvers to compute.

### 7.1 Helmholtz Perfectly Matched Layer Formulation

To solve the Helmholtz equation uniquely over a finite domain, we use a perfectly matched layer formulation, which attenuates waves on the boundary of the domain. Following Chen et al. [13] we

rewrite the Helmholtz equation as

$$\frac{\partial}{\partial x_1}\left(\frac{e_{x_2}}{e_{x_1}}\frac{\partial \Phi(\mathbf{x})}{\partial x_1}\right) + \frac{\partial}{\partial x_2}\left(\frac{e_{x_1}}{e_{x_2}}\frac{\partial \Phi(\mathbf{x})}{\partial x_2}\right) + e_{x_1}e_{x_2}k^2\Phi(\mathbf{x}) = -f(\mathbf{x}) \tag{11}$$

where $\mathbf{x} = (x_1, x_2) \in \Omega$, $e_{x_i} = 1 - j\frac{\sigma_{x_i}}{\omega}$, $k = \omega/c$, and

$$\sigma_{x_i} = \begin{cases} a_0\omega\left(\dfrac{l_{x_i}}{L_{\mathrm{PML}}}\right)^2 & x_i \in \partial\Omega \\[3mm] a_0\omega\left(\dfrac{l_{x_i}}{L_{\mathrm{PML}}}\right)^2 & \text{else} \end{cases}.$$

Here, $a_0$ controls the amount of wave attenuation (we use $a_0 = 5$), $l_{x_i}$ is the distance to the PML boundary along the $x_i$ axis, and $L_{\mathrm{PML}}$ is the width of the PML. Note that the PML is applied only to the boundary of our domain $\partial\Omega = \{x \,|\, 0.5 < \|x\|_\infty < 1\}$ and that the equation is identical to the original Helmholtz equation elsewhere. To train SIREN, we optimize Eq. 11 using the loss function described in the main paper with $\lambda(\mathbf{x}) = k = $ batch size$/5 \times 10^3$.

## 7.2 Full-Waveform Inversion

For known source positions and sparsely sampled wavefields, full-waveform inversion (FWI) can be used to jointly recover the wavefield and squared slowness over the entire domain. Specifically, FWI involves solving the constrained partial differential equation

$$\underset{m,\Phi}{\arg\min} \sum_{1 \le i \le N} \int_\Omega |\mathrm{III}_r(\Phi_i(\mathbf{x}) - r_i(\mathbf{x}))|^2 \, d\mathbf{x} \quad \text{s.t.} \quad H(m)\,\Phi_i(\mathbf{x}) = -f_i(x) \quad 1 \le i \le N, \forall \mathbf{x} \in \Omega,$$
$$\tag{12}$$

where there are $N$ sources, $\mathrm{III}_r$ samples the wavefield at the receiver locations, and $r_i$ is the measured receiver data for the $i$th source.

We solve this equation with a principled method based on the alternating direction method of multipliers [11, 1]. To perform FWI with SIREN, we first pre-train the network to solve for the wavefields given a uniform velocity model. This is consistent with the principled solver, which is initialized with a uniform velocity. This pre-training process updates SIREN to minimize

$$\mathcal{L}_{\mathrm{FWI, pretrain}} = \mathcal{L}_{\mathrm{Helmholtz}} + \lambda_{\mathrm{slowness}}\mathcal{L}_{\mathrm{slowness}} \tag{13}$$

where the first term is as described in the main paper, and the last term is simply $\|m(\mathbf{x}) - m_0\|_1$. $m(\mathbf{x})$ is parameterized using a single output of SIREN and we use an initial squared slowness value of $m_0 = 1$ in our experiments. The loss term $\mathcal{L}_{\mathrm{slowness}}$ is calculated over all sampled locations $\mathbf{x}$ in each minibatch. We also parameterize the multiple wavefields with additional SIREN outputs. This is accommodated in the loss function by sampling all source locations at each optimization iteration and applying the loss function to the corresponding wavefield outputs. Finally, we set $k = $ batch size$/10^4$ and $\lambda_{\mathrm{slowness}} = $ batch size.

After pre-training, we perform FWI using SIREN with a penalty method variation [44] of Eq. 12 as a loss function. This is formulated as

$$\mathcal{L}_{\mathrm{FWI}} = \mathcal{L}_{\mathrm{Helmholtz}} + \lambda_{\mathrm{data}}\mathcal{L}_{\mathrm{data}} \tag{14}$$

where $\mathcal{L}_{\mathrm{data}} = \sum_i \|\Phi_i(\mathbf{x}) - r_i(x)\|_1\Big|_{x \in \Omega_r}$, and $\Omega_r$ is the set of receiver coordinates. In other words, we add a weighted loss term using the (PML) Helmholtz formulation on the receiver coordinates. Here we use the same values of $k$ and $\lambda_{\mathrm{slowness}}$ as for pre-training.

## 7.3 Helmholtz Implementation & Reproducibility Details

**Data.** The dataset consists of randomly sampled coordinates and a Gaussian source function, as described previously. For neural FWI, the data term of the loss function uses the sampled wavefield values from the output of the principled solver using the same source and receiver locations, but with access to the ground truth velocity.

**Architecture.** For all Helmholtz experiments, the SIREN architecture (and baselines) use 5 layers with a hidden layer size of 256.

**Hyperparameters.** We set the loss function hyperparameters to the previously described values in order to make each component of the loss approximately equal during the beginning of training. The Adam optimizer with a learning rate of $2 \times 10^{-5}$ was used for all experiments.

**Runtime.** The single-source Helmholtz experiments were trained for 50,000 iterations requiring approximately 3 hours (ReLU), 8 hours (tanh, SIREN), or 24 hours (RBF). For FWI, pretraining required 80,000 (22 hours) iterations in order to suitably fit the 5 wavefields with a single network, and then we performed full-waveform inversion for 10,000 iterations (5 hours) until the loss appeared to converge. We set the batch size to fill the GPU memory; generally, we found that large batch sizes ranging from 3000 to 13000 samples worked well.

**Hardware.** The experiments are conducted on a NVIDIA Quadro RTX 6000 GPU (24 GB of memory).

### 7.4 Wave Equation Formulation

The wave equation is given by

$$\frac{\partial^2 \Phi}{\partial t^2} - c^2 \frac{\partial^2 \Phi}{\partial \mathbf{x}^2} = 0. \tag{15}$$

Note that in contrast to the Helmholtz equation, the wave equation is dependent on time. Thus, we parameterize the real-valued wavefield as a function of two spatial dimensions and time: $\Phi(t, \mathbf{x})$. We are interested in solving an initial value problem with the following initial conditions

$$\frac{\partial \Phi(0, \mathbf{x})}{\partial t} = 0 \tag{16}$$

$$\Phi(0, \mathbf{x}) = f(\mathbf{x}). \tag{17}$$

In the case of the acoustic wave equation, the first condition states that the initial particle velocity is zero, and in the second condition, $f(\mathbf{x})$ is an initial pressure distribution.

### 7.5 Solving the Wave Equation

We solve the wave equation by parameterizing $\Phi(t, \mathbf{x})$ with SIREN. Training is performed on randomly sampled points $\mathbf{x}$ within the domain $\Omega = \{\mathbf{x} \in \mathbb{R}^2 \,|\, \|\mathbf{x}\|_\infty < 1\}$. The network is supervised using a loss function based on the wave equation:

$$\mathcal{L}_{\text{wave}} = \int_\Omega \left\| \frac{\partial^2 \Phi}{\partial t^2} - c^2 \frac{\partial^2 \Phi}{\partial \mathbf{x}^2} \right\|_1 + \lambda_1(\mathbf{x}) \left\| \frac{\partial \Phi}{\partial t} \right\|_1 + \lambda_2(\mathbf{x}) \left\| \Phi - f(\mathbf{x}) \right\|_1 \, d\mathbf{x}. \tag{18}$$

Here, $\lambda_1$ and $\lambda_2$ are hyperparameters, and are non-zero only for $t = 0$. To train the network, we sample values of $\mathbf{x}$ uniformly from $\Omega$ and slowly increase the value of $t$ linearly as training progresses, starting from zero. This allows the initial condition to slowly propagate to increasing time values. We set $\lambda_1 = \text{batch size}/100$ and $\lambda_2 = \text{batch size}/10$ and let $c = 1$.

Results are shown in Fig. 6 for solving the wave equation with $f(\mathbf{x})$ equal to a Gaussian centered at the origin with a variance of $5 \times 10^{-4}$. We also compare to a baseline network with tanh activations (similar to recent work on neural PDE solvers [38]), and additional visualizations are shown in the video. SIREN achieves a solution that is close to that of a principled solver [41] while the tanh network fails to converge to a meaningful result.

### 7.6 Wave Equation Implementation & Reproducibility Details

**Data.** The dataset is composed of randomly sampled coordinates 3D coordinates as described previously. We use a Gaussian source function to approximate a point source, and clip the support to values greater than 1e-5. During training, we scale the maximum value of the Gaussian to 0.02, which we find improves convergence.

Figure 6: Solving the wave equation initial value problem. For an initial condition corresponding to a Gaussian pulse, SIREN recovers a wavefield that corresponds closely to a ground truth wavefield computed using a principled wave solver [41]. A similar network using tanh activations fails to converge to a good solution. MSE values are shown for each frame, where the time value is indicated in the top row.

**Architecture.**  To fit over the 3 dimensions of the wave equation, we increase the size of the architecture, still using 5 layers, but with a hidden layer size of 512.

**Hyperparameters.**  The loss function hyperparameters are set so that each component of the loss is approximately equal as training progresses. We grow the interval of $t$ from which training coordinates are sampled linearly over 100,000 iterations (roughly 25 hours) from 0.0 to 0.4, which we find allows a sufficient number of iterations for the network to fit the wave function as it expands. For all wave equation experiments, we used the ADAM optimizer and a learning rate of $2 \times 10^{-5}$. A batch size of 115,000 is used, which fills the GPU memory.

**Hardware.**  The experiments are conducted on a NVIDIA Quadro RTX 6000 GPU (24 GB of memory).

## 8  Application to Image Processing

### 8.1  Formulation

As shown previously, one example of signal that SIRENs can be used to represent are natural images. A continuous representation of natural images with a SIREN introduces a new way to approach image processing tasks and inverse problems. Consider a mapping from continuous implicit image representation $\Phi(x, y)$ to discrete image $b$

$$b = \text{III} \left( h * \Phi_\theta \left( x, y \right) \right), \tag{19}$$

where III is the sampling sampling operator, $h$ is a downsampling filter kernel, and $\Phi(x, y)$ is the continuous implicit image representation defined by its parameters $\theta$. Using this relationship, we can fit a continuous SIREN representation given a discrete natural image $b$ by supervising on the sampled discrete image.

Many image processing problems can be solved by formulating an optimization problem which minimizes data fidelity with partial or noisy measurements of $b$ and some prior over natural images. In our case, our prior is over the space of SIREN representations of natural images. This takes the form:

$$\underset{\{\theta\}}{\text{minimize}} \ \mathcal{L} \left( \text{III} \left( h * \Phi_\theta \left( x, y \right) \right), b \right) + \lambda \gamma \left( \Phi_\theta \left( x, y \right) \right), \tag{20}$$

Figure 7: Comparison of different implicit network architectures fitting a ground truth image (top left). The representation is only supervised on the target image but we also show first- and second-order derivatives of the function fit in rows 2 and 3, respectively. We compare with architectures implemented using Softplus, ELU, SELU, and ReLU P.E. (L=4) on the cameraman image. The value of L dictates the number of positional encodings concatenated for each input coordinate, and a choice of $L = 4$ was made for images in [36].

where $\gamma$ is a regularization function defined on the continuous function, and $\lambda$ is the weight of the regularizer.

## 8.2 Image Fitting.

As previously shown, the most simple representation task involves simply fitting an implicit neural representation $\Phi : \mathbb{R}^2 \mapsto \mathbb{R}^3, \mathbf{x} \to \Phi(\mathbf{x})$ to an image. Simply fitting the image proves to be challenging for many architectures, and fitting higher-order derivatives is only possible using SIRENs. In addition to the comparisons with ReLU, tanh, ReLU P.E., and ReLU with RBF input layer shown in the paper, we show a qualitative comparison with additional neural network architectures in Fig. 7.

## 8.3 Image Inpainting

Traditional approaches to the single image inpainting task have either focused on diffusion-based [8, 5, 7] or patch based reconstruction [6, 25, 15]. With the advent of deep learning, a slew of new methods have taken advantage of large amounts of data to learn complex statistics of natural images used in reconstruction problems. These inpainting methods are based on convolutional neural networks (CNNs) [28, 39] and generative adversarial networks (GANs) [22, 45, 29, 33]. Additionally, neural network architectures for image recovery like CNNs have been shown to themselves act as a prior [42] for natural images, allowing for solving inverse problems without the use of training data.

We show the capability of SIRENs to solve inverse problems through the example of single image inpainting. By fitting a SIREN to an image and enforcing a prior on the representation, we can solve a single image reconstruction problem. Examples of single image inpainting with and without priors are shown in Fig. 8, where we compare performance on texture images versus Deep Image Prior [42], Navier-Stokes, Fluid Dynamics Image Inpainting [7] (Diffusion), and SIRENs with no prior, total variation prior (TV), and Frobenius norm of Hessian [32, 31] priors (FH) respectively. In Tab. 3, we describe our quantitative results with mean and standard deviation over many independent runs. These results show that SIREN representations can be used to achieve comparable performance to other baseline methods for image inverse problems.

Note that this formulation of loss function can be equivalently formulated in a continuous partial differential equation, and depending on choice of prior, a diffusion based update rule can be derived. For more details on this, see the Rudin–Osher–Fatemi model in image processing [4, 3, 17].

Figure 8: Comparison of various methods and priors on the single image inpainting task. We sample 10% of pixels from the ground truth image for training, learning a representation which can inpaint the missing values. Note that for the image in the first row, where the TV prior is known to be accurate, including the TV prior improves inpainting performance.

## 8.4 Implementation & Reproducibility Details

**Data.** The experiments were run on texture images, including the art image of resolution $513 \times 513$ and tiles image of resolution $355 \times 533 \times 3$. These images will be made publicly available with our code. The sampling mask is generated randomly, with an average of 10% of pixels being sampled. We will make the example mask for which these results were generated publicly available with our code. As in all other applications, the image coordinates $\mathbf{x} \in \mathbb{R}^2$ are normalized to be in the range of $[-1, 1]^2$. For evaluation, images are scaled in the range of $[0, 1]$ and larger values are clipped.

**Architectures.** For the single image inpainting task with SIRENs, we use a 5-layer MLP. For single image fitting on the cameraman image, we use 5-layer MLPs for all activation functions. For the RBF-Input and ReLU P.E. models, we add an additional first layer with 256 activations (in the case of RBF-Input) or positional encoding concatenation with positional encoding sinusoid frequencies of $2^i \pi$ for $0 \le i < L = 7$ (in the case of ReLU P.E.).

**Loss Functions.** In order to evaluate a prior loss, we must enforce some condition on the higher-order derivatives of the SIREN. This is done by sampling $N$ random points $\mathbf{x}_i \in [-1, 1]^2$, and enforcing the prior on these points. We sample half as many points for the prior as there are pixels in the image. In the case of TV regularization, this consists of a L1 norm on the gradient

$$\gamma_{\text{TV}} = \frac{1}{N} \sum_{i=1}^{N} |\nabla \Phi_\theta(\mathbf{x}_i)|, \tag{21}$$

while in the case of FH regularization, this consists of L1 norm on all sampled points' Frobenius norm of their Hessian matrix

$$\gamma_{\text{FH}} = \frac{1}{N} \sum_{i=1}^{N} \|\text{Hess}(\Phi_\theta(\mathbf{x}_i))\|_F. \tag{22}$$

The prior loss is weighted with a regularization weight $\lambda$, and combined with the MSE loss on the reconstructed sampled and blurred image points,

$$\mathcal{L}_{\text{img}} = \|\text{Ш} (h * \Phi_\theta(x, y)) - b\|^2. \tag{23}$$

**Downsampling Kernel Implementation.** Sampling images from a continuous function requires convolution with a downsampling kernel to blur high frequencies and prevent aliasing. Since we cannot perform a continuous convolution on a SIREN we must instead approximate with Monte Carlo sampling of the SIREN to approximate fitting the blurred function. Consider the 2D image signal where $\mathbf{x} = (x, y)$:

$$(h * \Phi)(x, y) = \int_{x'} \int_{y'} \Phi(x', y') \cdot h(x - x', y - y') dy' dx' \approx \frac{1}{N} \sum_{i=1}^{N} \Phi(x + x_i, y + y_i) \tag{24}$$

Table 3: Mean and standard deviation of the PSNR of the tiles texture and art texture images for SIRENs with various priors. The statistics are computed over 10 independent runs.

| Image | No Prior Mean PSNR | No Prior Std. PSNR | TV Prior Mean PSNR | TV Prior Std. PSNR | FH Prior Mean PSNR | FH Prior Std. PSNR |
|---|---|---|---|---|---|---|
| Tiles | 15.45 | 0.180 | 17.40 | 0.036 | 17.68 | 0.051 |
| Art | 32.41 | 0.283 | 34.44 | 0.222 | 27.18 | 0.116 |

Figure 9: Example frames from fitting a video with SIREN and ReLU MLPs. SIREN more accurately reconstructs fine details in the video. Mean (and standard deviation) of the PSNR over all frames is reported.

where $x_i, y_i$ are sampled from the kernel $h$ as a normalized probability density function. For example, a bilinear downsampling kernel is given by $h(x, y) = \max(0, 1 - |x|) \max(0, 1 - |y|)$. Thus, we sample $x_i, y_i$ from a probability density function of $p(x_i, y_i) = \frac{1}{2} \max(0, 1 - |x|) \max(0, 1 - |y|)$. In our implementation, we found that not using a downsampling kernel resulted in equivalent performance on the inpainting and image fitting task. However, it may be necessary in cases where we aim to reconstruct our image at multiple resolutions (i.e. superresolution). We only sample one blurred point, i.e. $N = 1$, per iteration and train for many iterations. This is done for computational efficiency, as otherwise it is necessary to backpropagate the loss from all sampled coordinates.

**Hyperparameters.** For the image fitting experiment, we train all architectures using the Adam optimizer and a learning rate of $1 \times 10^{-4}$. Hyperparameters were not rigorously optimized and were found by random experimentation in the range of $[1 \times 10^{-6}, 1 \times 10^{-4}]$. We train for 15,000 iterations, fitting all pixel values at each iteration.

For the image inpainting experiments, we use the published and OpenCV [12] implementations for the baseline methods, and use an Adam optimizer with a learning rate of $5 \times 10^{-5}$ for all SIREN methods. We train for 5,000 iterations, fitting all pixel values at each iteration. For the TV prior, we use a regularization weight of $\lambda = 1 \times 10^{-4}$, while for the FH prior, we use a regularization weight of $\lambda = 1 \times 10^{-6}$.

**Central Tendencies of Metrics.** In Tab. 3, we show the central tendencies (mean and standard deviation) of the quantitative PSNR scores obtained on the image inpainting experiment. Inpainting with SIRENs is highly stable and not sensitive to the specific pixel mask sampled.

**Hardware & Runtime.** We run all experiments on a NVIDIA Quadro RTX 6000 GPU (24 GB of memory). The single image fitting and regularization experiments require approximately 1 hour to run.

## 9 Representing Video

We fit videos using SIREN and a baseline ReLU architecture as described in the main paper and video. We also fit a second video, which consists of various vehicles moving in traffic and outdoor scenes, shown in Fig. 9. Again, SIREN shows improved representation of fine details in the scene. In the following we provide additional implementation details.

### 9.1 Reproducibility & Implementation Details

**Data.** The first dataset consists of a video of a cat, which is permissively licensed and available at the time of this writing from `https://www.pexels.com/video/the-full-facial-features-of-a-pet-cat-3040808/`. The second dataset is the "bikes sequence" available from the scikit-video Python package described here `http://www.scikit-video.org/stable/datasets.html`. We crop and downsample the cat video to 300 frames of $512 \times 512$ resolution. The second dataset consists of 250 frames fit at the original resolution of $272 \times 640$ pixels.

**Architecture.** The SIREN and ReLU architectures use 5 layers with a hidden layer size of 1024.

**Hyperparameters.** The Adam optimizer with a learning rate of $1 \times 10^{-4}$ was used for all experiments. We set the batch size to fill the memory of the GPUs (roughly 160,000).

**Runtime.** We train the videos for 100,000 iterations, requiring approximately 15 hours.

**Hardware.** The networks are trained using NVIDIA Titan X (Pascal) GPUs with 12 GB of memory.

## 10 Representing Audio Signals

Various methods exist for audio signal representation. Early work consists of representing audio signals using various spectral features [26, 9, 24]. Spectrograms, representations of the spectrum of frequencies of a signal as it varies with time, have been used in machine learning applications due to the ease of applying widely successful image processing CNN architectures to them [21, 14, 40]. More recently, neural network architectures have been developed which can operate on raw audio waveforms [43, 14, 35].

To demonstrate the versatility of SIRENs as implicit neural representations, we show that they can efficiently model audio signals. Due to the highly periodic nature of audio signals with structure at various time scales, we expect that SIRENs could accurately represent such signals efficiently and provide an alternative representation for audio signals. We evaluate SIREN performance on raw audio waveforms of varying length clips of music and speech. While other neural network architectures fail to accurately model waveforms, SIRENs are able to quickly converge to a representation which can be replayed with minimal distortion. We fit a SIREN to a sampled waveform $a$ using a loss of the form:

$$\mathcal{L} = \int_{\Omega} \|\text{Ш}_a(\Phi(\mathbf{x})) - a(\mathbf{x})\|^2 \, d\mathbf{x}. \tag{25}$$

where $\text{Ш}_a$ samples the SIREN at the waveform measurement locations.

Fig. 10 displays the fit waveform to music and speech data respectively. We see that other neural network architectures are not able to represent raw audio waveforms at all, while SIRENs produce an accurate waveform. Additionally, we note that the number of parameters in out SIREN is far less than the number of samples in the ground truth waveform. This ability to compress signals supports our claim that periodic SIREN representations are well suited to representing audio signals, and perhaps lossy compression algorithms for audio could be designed using SIRENs. Our supplemental video contains audio from the SIREN, which is accurate and recognizable. Tab. 4 shows the converged SIREN mean-squared error on the original audio signal and statistics on these metrics (these were feasible to evaluate due to the relatively short training time of SIRENs on audio signals). This shows SIRENs are highly stable in convergence.

### 10.1 Reproducibility & Implementation Details

**Data.** For music data, we use the first 7 seconds from Bach's Cello Suite No. 1: Prelude available at `https://www.yourclassical.org/story/2017/04/04/daily-download-js-bach--cello-suite-no-1-prelude` and for the speech we use stock audio of a male actor counting from 0 to 9 available at `http://soundbible.com/2008-0-9-Male-Vocalized.html`. These waveforms are have a sampling rate of 44100 samples

Figure 10: Fitted waveforms and error for various implicit neural representation architectures. We fit the network to the first 7 seconds of Bach's Cello Suite No. 1: Prelude (Bach) and to a 12 second clip of a male actor counting 0-9 (Counting). Only SIREN representations capture the waveform structure.

Table 4: Mean squared error of representing the raw audio waveform scaled in the range $[-1, 1]$ with a SIREN. The mean and variance of the reconstruction MSE are evaluated over 10 independent runs of fitting. Each architecture is fitted for 5000 iterations.

| Architecture | Bach MSE Mean | Bach MSE Standard Dev. | Counting MSE Mean | Counting MSE Standard Dev. |
|---|---|---|---|---|
| ReLU | $2.504 \times 10^{-2}$ | $1.706 \times 10^{-3}$ | $7.466 \times 10^{-3}$ | $8.217 \times 10^{-5}$ |
| ReLU P.E. | $2.380 \times 10^{-2}$ | $3.946 \times 10^{-4}$ | $9.078 \times 10^{-3}$ | $9.627 \times 10^{-4}$ |
| SIREN | $1.101 \times 10^{-5}$ | $2.539 \times 10^{-6}$ | $3.816 \times 10^{-4}$ | $1.632 \times 10^{-5}$ |

per second. As pre-processing, they are normalized to be in the range of $[-1, 1]$. We use the entire set of samples to fit our SIREN in each batch.

**Architecture.** We use the same 5-layer MLP with sine nonlinearities as for all other SIREN applications.

**Frequency Scaling.** To account for the high sampling rate of audio signals, we scale the domain $\mathbf{x} \in [-100, 100]$ instead of $[-1, 1]$. This is equivalent to adding a constant multiplication term to the weights of the input layer of the SIREN.

**Hyperparameters.** We use the Adam optimizer with a learning rate of $5 \times 10^{-5}$ to generate the results. We evaluated both learning rates of $5 \times 10^{-5}$ and $1 \times 10^{-4}$, finding that $5 \times 10^{-5}$ worked slightly better. We train for 9,000 iterations for the figures generated, and 5,000 iterations for the quantitative results listen in the table (the model is largely converged after only 2,000 iterations).

**Hardware & Runtime.** The experiments are conducted on a NVIDIA Quadro RTX 6000 GPU (24 GB of memory), where training for 9000 iterations takes roughly 20 minutes for the Bach signal and 30 minutes for the counting signal.

## 11 Learning a Space of Implicit Functions

A strong prior over the space of SIREN functions enables applications such as reconstruction from noisy or few observations. We demonstrate that this can be done over the function space of SIRENs

representing faces in the CelebA dataset [30]. We use the learned prior to perform image inpainting of missing pixels.

## 11.1 Reproducibility & Implementation Details

**Data.** Partial observations (referred to as context) of the input image consist of coordinates and pixel values $O = \{(x_i, c_i)\}_{i=0}^{N}$ sampled from an image $b \in \mathbb{R}^{H \times W \times 3}$. Like in [16], $b \in \mathbb{R}^{32 \times 32 \times 3}$ is center-cropped and downsampled from the images in the CelebA training dataset, containing 162,770 images. We evaluate our test performance on a similarly center-cropped and downsampled version of the CelebA test dataset, containing 19,962 images.

**Context Encoder.** The results presented in the main paper use a convolutional neural network encoder which operates on sparse images. More specifically, the partial observations are combined into sparse images $O \in \mathbb{R}^{32 \times 32 \times 3}$, where observed pixel locations are either their value $c_i$ and masked pixel locations are given a value of $0$. The encoder $C$ operates on these sparse images, and is parameterized as a standard convolutional neural network (CNN) with an input convolutional layer followed by four residual blocks with ReLU nonlinearities. Each intermediate feature map has 256 channels. This outputs per-pixel embeddings in $\mathbb{R}^{256}$, which are aggregated together into a single context embedding using a fully connected layer.

We also describe the use of a set encoder as in [16] for encoding the partial observations. In this case, partial observations consist of a list of coordinates and pixel values $O = \{(x_i, c_i)\}_{i=0}^{N}, (x_i, c_i) \in \mathbb{R}^5$. The encoder $C$ is an MLP which operates on each of these observations independently. The MLP consists of two hidden layers with sine nonlinearities, and outputs an embedding per pixel in $\mathbb{R}^{256}$. The embeddings are aggregated together using a mean operation. Since each embedding depends only on the context pixel, and the mean operation is symmetric, this set encoder is permutation invariant.

We consider one final encoder $C$ based on partial convolutions [28]. Partial convolutions are designed to operate on sparse images, conditioning outputs of each layer only on valid input pixels. In this case, the partial observations are combined into a sparse image and mask, much like in the CNN encoder case. However, the encoder is implemented using an input partial convolution followed by four partial convolution residual layers with ReLU nonlinearities. Each intermediate map also has 256 channels. The output per-pixel embeddings are similarly aggregated togheter into a single context embedding using a fully connected layer.

**Hypernetwork.** We use a hypernetwork as our decoder, which maps the latent code to the weights of a 5-layer SIREN with hidden features of size 256 (as in all other experiments). This hypernetwork is a ReLU MLP with one hidden layer with 256 hidden features.

**Loss Function.** We train the encoder $C$ and hypernetwork $\Psi$ operating on context $O$ by minimizing the loss function:

$$\mathcal{L} = \underbrace{\frac{1}{HW} \|\Phi(\mathbf{x}) - b\|_2^2}_{\mathcal{L}_{\text{img}}} + \lambda_1 \underbrace{\frac{1}{k} \|z\|_2^2}_{\mathcal{L}_{\text{latent}}} + \lambda_2 \underbrace{\frac{1}{l} \|\theta\|_2^2}_{\mathcal{L}_{\text{weights}}} \tag{26}$$

where $(H, W)$ are the spatial dimensions of the images in the dataset, $\Phi = (\Psi \circ C)(O)$ is the predicted SIREN representation from the hypernetwork, $b$ is the ground truth image, $k$ is the dimensionality of the embedding $z$, and $l$ is the amount of weights $\Phi$ in the SIREN $\Phi$.

$\mathcal{L}_{\text{img}}$ enforces the closeness of image represented by the SIREN to ground-truth, $\mathcal{L}_{\text{latent}}$ enforces a Gaussian prior on latent code $z$, and $\mathcal{L}_{\text{weights}}$ is a regularization term on the weights of $\Phi$ which can be interpreted as encouraging a lower frequency representation of the image. The regularization terms are necessary since there are many possible SIREN representations for an image, so we need to encourage unique solutions (lowest possible frequency) which lie in a more compact latent space (Gaussian). For all of our results, we use regularization weighting parameters of $\lambda_1 = 1 \times 10^{-1}$ and $\lambda_2 = 1 \times 10^2$.

**Hypernetwork Initialization.** In order to improve performance, we devise a heuristic initialization scheme for the hypernetwork which deviates from the default Kaiming initialization for ReLU MLP networks [20]. Although a formal theoretical analysis of this initialization has not been well studied,

Table 5: Quantitative comparison of inpainting on the CelebA test dataset. Metrics are reported in pixel-wise mean squared error for varying numbers of context pixels. All of the methods for generalizing over SIRENs use a hypernetwork as a decoder from latent code to SIREN weights. CNP does not report quantitative metrics on half or full images given as context.

| Number of Context Pixels | 10 | 100 | 1000 | 512 (Half) | 1024 |
|---|---|---|---|---|---|
| CNP [16] | 0.039 | 0.016 | 0.009 | - | - |
| Sine Set Encoder + Hypernet. | 0.035 | 0.013 | 0.009 | 0.022 | 0.009 |
| ReLU Set Encoder + Hypernet. | 0.040 | 0.018 | 0.012 | 0.026 | 0.012 |
| PConv CNN Encoder + Hypernet. | 0.046 | 0.020 | 0.018 | 0.060 | 0.019 |
| CNN Encoder + Hypernet. | **0.033** | **0.009** | **0.008** | **0.020** | **0.008** |

we found that the initialization led to convergence of our encoder and hypernetwork models. We only modify the default ReLU MLP initialization in the final layer of the hypernetwork by scaling the Kaiming initialized weights by $1 \times 10^{-2}$, and initializing the biases uniformly in the range of $[-1/n, 1/n]$ where $n$ is the number of inputs to the layer of the SIREN being predicted.

The motivation for this scheme is that the initialization of the biases of the hypernetwork is a heuristic initialization of SIRENs which leads to high quality convergence results. Thus, initializing the weights of the hypernetwork with a small magnitude ensures that the SIREN weights outputted at initialization of the hypernetwork are close to a initialization of a single SIREN, regardless of input to the hypernetwork.

**Training Procedure.** In order to encourage invariance to the number of partial observations, we randomly sample from 10 to 1000 context pixels to input into the convolutional or partial convolutional encoder. In the case of the set encoder which is permutation invariant, we mimic the training procedure of [16] by varying from 10 to 200 sampled context pixels.

**Hyperparameters.** As mentioned, we use loss parameters $\lambda_1 = 1 \times 10^{-1}$ and $\lambda_2 = 1 \times 10^2$. For all experiments, we use the Adam optimizer with a learning rate of $5 \times 10^{-5}$, a batch size of 200 images, and train for 175 epochs on the training dataset. We found these hyperparameters by trial and error, having tested values of $\lambda_1 \in [10^{-3}, 10^{-1}]$, $\lambda_2 \in [10^1, 10^4]$, learning rates of $5 \times 10^{-5}, 1 \times 10^{-4}$, and a batch size of 200 and 1000.

**Runtime.** We train the videos for 175 epochs on the downsampled CelebA training set, requiring approximately 24 hours.

**Hardware.** The networks are trained using NVIDIA Quadro RTX 6000 GPUs with 24 GB of memory.

## 11.2 Additional Results

We show additional results from the convolutional encoder in Fig. 11

We also show results from the set encoder with sine nonlinearities in Fig. 12, set encoder with ReLU nonlinearities (as in the original CNP architecture) in Fig. 13, and convolutional encoder based on partial convolutions in Fig. 14. All of these implementations use the same hypernetwork architecture as a decoder from latent codes to SIREN weights. Tab. 5 shows comparisons between architectures for the encoder.

Interestingly, the partial convolutional encoder performs worse than both the set encoders and convolutional encoder. We suspect that the convolutional encoder has an easier time capturing complex spatial relationships between the context pixels and using information from the masked pixels instead of only conditioning on valid pixels. Regardless of encoder architecture, some prior over the space of SIRENs has been learned which can be used to perform inpainting comparably to methods such as CNP [16] operating on images directly.

Figure 11: Additional results using the CNN encoder with hypernetwork decoder.

Figure 12: Additional results using the set encoder with sine nonlinearities with a hypernetwork decoder.

Figure 13: Additional results using the set encoder with ReLU nonlinearities with a hypernetwork decoder.

Figure 14: Additional results using a CNN with partial convolution encoder with a hypernetwork decoder.