[Reviews · NeurIPS 2020]

Review 1

Summary and Contributions: This is a very interesting, ambitious, insightful and solid piece of work that I find very impressive. The authors proposed a novel class of implicit function representations using MLPs that leverage sinusoidal activation functions. They analytically worked out an initialization scheme for the weights and biases of such networks, which exhibits superior convergence properties. They further illustrate the capabilities of such networks on a sweeping range of applications, from image/audio/video representation to solving various classic partial differential equations such as the linear Poisson and Helmholtz equations, as well as the nonlinear Eikonal equation, achieving impressive results.

Strengths: There are various aspects of this paper that I find to be very strong: (1) First and foremost, implicit function representations using MLPs is a trending topic in machine learning that is pushing various frontiers including 3D geometric deep learning, with many potential applications in surrounding fields. This study will certainly offer the prospect of extending this line of research into many other fields in computer science, for applications such as generative modeling of image/video/speech, etc. (2) The main insight of this method is very simple, yet ex-post obvious. Exploiting periodic activations will allow the network to better model signals at a range of dramatically different frequencies.The mathematical derivations (as detailed in the appendices, though I have not gone through every equation thoroughly) are comprehensive and detailed. (3) The breadth of the research is remarkable, covering a wide range of applications in computer science.

Weaknesses: Overall I am very pleased with the current presentation of the study. A few points to consider: (1) The breadth of the experiments are incredible, but perhaps most experiments involve somewhat trivial tasks such as "fitting an image" or "fitting a function" (though I understand the objective of the experiments is to illustrate representational power of these SIREN networks). The Space of Implicit Functions nicely ties the method to prior art that learns a prior over training datasets, but comparisons are rather rudimentary. (2) Though this is not a call for additional experimental evidence, I am rather surprised that for the Poisson Equation experiment, the authors opted for the more trivial example of fitting an image by its derivatives, rather than opting for the more useful Poisson Surface Reconstruction task (fitting 3d implicit fields given gradient samples from surface points and normals). Might be an application to consider given this methodology. (3) This is more of a question than a weakness, but I can't find a more suitable place to put this. For solving PDEs, just conforming to the source function spatially (temporally), is insufficient, since there needs to be a boundary condition for boundary value problems (BVPs), and boundary + initial conditions for initial value problems (IVPs). Without boundary conditions, the solutions could be arbitrary. For the Poisson Equation example, as well as the Helmholtz Equation example, I don't see a set of boundary conditions being posed (and a corresponding penalty function for enforcing them). Am I missing something important?

Correctness: The methodologies of the paper are correct in my professional opinion. The evaluations are comprehensive, mathematical derivations are thorough, and reproducibility materials are sufficient.

Clarity: The paper is very well written, very clearly stating the main methodological contributions (periodic activation for implicit networks, and principled activation methods), and the range of target applications.

Relation to Prior Work: The literature review section has been quite comprehensive. I would personally be interested if the authors could offer some insights into the similarities with the classic Spectral Methods for solving PDEs with Fourier bases. I am pretty excited about the prospect of connecting the traditional / DL ways of solving PDEs. This work bears many similarities with a concurrent work: [1] Tancik, Matthew, et al. "Fourier Features Let Networks Learn High Frequency Functions in Low Dimensional Domains." arXiv preprint arXiv:2006.10739 (2020). Another concurrent work on extending learned implicit function representations towards solving general PDEs: [2] Jiang, Chiyu Max, et al. "MeshfreeFlowNet: A Physics-Constrained Deep Continuous Space-Time Super-Resolution Framework." arXiv preprint arXiv:2005.01463 (2020).

Reproducibility: Yes

Additional Feedback: N/A


Review 2

Summary and Contributions: Authors propose a method for learning implicit representations using neural networks with sinusoidal activation functions. In short, given an implicit function on some continuous domain (e.g. spatial and / or temporal coordinates), authors propose approximating this function with a multi-layer perceptron with a sinus as non-linearities, and propose a principled way for initializing the weights of the network. The work is validated on a diverse set of applications, which include learning shape representations, audio and video signals, solving Poisson equations, and others.

Strengths: + The core idea makes a lot of sense and is very simple to implement. + The problem of learning implicit functions itself is very important, as there are a lot of applications (e.g. learning 3D shape representations) which can directly benefit from this approach. + The range of presented applications is very diverse, which indicates that the method will be useful for many practitioners. + Evaluation is very broad, and in particular, the visual results on learning SDFs seem quite impressive (although see weaknesses).

Weaknesses: - Authors introduce formulations for a wide range of applications, yet the numerical evaluation is not very elaborate: for example, I am not sure why they did not report numerical results on learning SDFs. - (minor) [5] work proposed a very similar idea (albeit it is less principled). Empirical comparison to that method is not there. - (minor) Although authors do propose a principled way to initialize the model (and provide an elaborate proof), it is not clear if this scheme will work when used with other common tricks e.g. batchnorm.

Correctness: The method and claims seem to be reasonable, there are no obvious mistakes. For some applications, the empirical evaluation is quite limited as there are only qualitative results, but it might be reasonable given how broad the number of tasks is.

Clarity: The paper is easy to follow, and is well-written. There are also enough details to reproduce the work.

Relation to Prior Work: The closest recent work is probably positional encodings (which is proposed in NERF [5]), and authors do discuss the differences with that work. However, since there is no experimental comparison, it is hard to understand if there are big benefits in practice over positional encodings (which seem even simpler to use - e.g. there is no need for any sort of special initialization). (minor) Wavelet transforms are not mentioned although the connection to those can be interesting.

Reproducibility: Yes

Additional Feedback: Authors provided a thorough rebuttal and tackled most of the reviewers concerns. All reviewers seem to agree that this is a good paper and it should be accepted.


Review 3

Summary and Contributions: This is a nice paper investigating the representation power of the network equipped with the periodic (e.g. sine) activation functions. Particularly, the fine details of functions to approximate would be largely preserved, and the converging speed would be up significantly, given the use of the periodic functions as the activation. One of the major advantage is that the value of the approximated functions can be predicted even only the first or second derivatives are provided (without the value itself provided). Multiple applications are provided as the examples to illustrate the power of the proposed periodic activation functions.

Strengths: + The paper is very well presented, carefully evaluated, and easy to follow. + The idea is simple but powerful. The redundancy of the network is significantly reduced when representing the same level of the function details. The learned network representation could be regarded as a high quality compressor. + The periodic activation function is general to many representations, including the pixel modeling, the 3D signed distance function, various physics scenarios (Poisson Equation / HelmHoltz Equations).

Weaknesses: - Analysis of the sensitivity of initialization - To my opinion, the sensitivity of the initialization of the sine activation function, seems more significant than existing activations, like the SoftPlus. When extending the model (e.g. from the current single field overfitting to code-to-field learning, where we need new initialization schemes), one might choose the approach with less sensitivity of initilizations for the extension attempt. It would be expected if the analysis of such sensitivity could be provided. For example, with the same deviation of the initialization statistics, which methods would achieve the minimum performance drop? - To me, setting the activation functions as the sine functions seem to be processing the signal in the frequency domain. One can pre-process the inputs (e.g. the x/y/z) with Fourier transform or z transform, and then being forwarded by the network with existing activation functions (e.g. softplus). I would be curious about the comparison between the proposed method and this baseline - mainly to uncover whether the high compression rate is from multiple layers' sine activations, or just because of processing the signals in the frequency domain.

Correctness: Correct.

Clarity: Very well written.

Relation to Prior Work: Yes.

Reproducibility: Yes

Additional Feedback:


Review 4

Summary and Contributions: This paper investigates how to use a sinusoidal activation in order to represent implicit functions. The paper finds that using such representations are extremely effective at representing high-frequency details for a variety of applications ranging from a simple reconstruction task of images and videos to learning functions using only their derivatives (e.g. solving the wave equation or performing poisson editing).

Strengths: Idea: I think that the ability to be able to model high-quality detail is a challenging problem -- it is especially important in generative applications to be able to generate high quality details and often current neural networks are somewhat lacking in this capacity without vast amounts of data. Significance: While others have worked on using periodic functions, the authors, I think, demonstrate how these networks can be used effectively for a variety of challenging, real world problems. They introduce a number of tricks (e.g. initialisation) and choose a domain for which these networks have a seemingly clear advantage -- implicit function representation. This allows them to do the following tasks to a much higher quality than a standard ReLU network: - image/video fitting - image editing based on the gradient - fitting a function (e.g. solving a pde) using the first or second deriv Range of experiments and quality of results: as mentioned in the paragraph above, the authors have demonstrated their results on a wide range of challenging, real world, realistic tasks which I think is impressive and demonstrates the utility of their approach.

Weaknesses: Idea: while the idea and results/implementation are clearly good, I was lacking some interpretation or intuition as to why the sinusoidal activation function is a good choice beyond the derivatives being well behaved. I think there were some interesting discussions in related work about inspiration from Fourier series and DCT for other work. I think it would be helpful in the intro to have some discussion of the authors' inspiration. Experiments: - the authors operate on a 5 layer network. While this allows for a range of realistic tasks, I wonder if training becomes more challenging as the layer increases in depth? It would be interesting then to explore layer depth vs training ease vs performance for other researchers. - Following up on the above point, how difficult were these models to train? Also did the authors consider different architectures -- one of the benefits of the ReLU setup is it works well for a variety of architectures.

Correctness: I believe so.

Clarity: Yes. While some more discussion would be useful (as mentioned above), in general the paper (and especially the video) are clear! There are a couple of aspects I was confused about though: - how do the authors train the models when they use only the derivatives? The paper seems to imply that they operate on the NN itself (the phi), as there is an extensive discussion about how the derivative of a SIREN is a SIREN again. However, I would have thought that it would have been simply implemented as the loss? It would be helpful if this were clarified. - for the video task: I assume there are four inputs: x, y, z, and t (where t is time?) Maybe it's worth putting somewhere the inputs for the different tasks

Relation to Prior Work: Yes.

Reproducibility: Yes

Additional Feedback: But obviously for reproducibility, the authors should release code. All the reviewers were in agreement that this work is great and should be accepted.

[Author Response · NeurIPS 2020]

We thank our reviewers for their positive comments, and the time they spent carefully reading our paper. Their feedback to clarify some aspects of our work and suggestions to perform additional experiments will improve our manuscript.

**Relation to prior work** We are glad to inaclude and discuss the papers **(R1,2)** suggested in our related work section. These manuscripts appeared at the time of our submission and could unfortunately not be discussed (Fourier Features arXiv, June'20 and MeshFreeFlowNet arXiv, May'20) but are indeed related to our work.

**Additional experiments (R2)** Qualitative & Quantitative comparisons with NERF's approach [5] do appear in the paper and suppl. under the name Positional Encoding (PE). Specifically, we compared our work to PE for image fitting (Fig.1.) and SDF (suppl.). In particular, we show in Fig.1 that PE does not yield sensible gradients and Laplacians despite fitting the image well. Those derivatives are crucial in our other experiments involving PDEs, and SIREN was the only architecture resulting in sensible higher order derivatives. **Quantitative experiments for SDF fitting (R2)** were performed using the Chamfer distance, eg. for the statue in a $[-1, 1]$ box, we obtain: ReLU=3.78e-4, PE=6.19e-5, SIREN=5.50e-5. We will add those new results to the paper. **Initialization Sensitivity (R3)** We trained SIREN and a Softplus-MLP on Poisson reconstruction for deviations of the initialization's std. dev. from 5% to 50%. SIREN outperforms the Softplus-MLP at all noise levels and is robust to changes in the std. dev. up to 20%, degrading only significantly at deviations of 50%. Further, we show that independent runs of SIRENs have low error std. dev. (suppl. Tab.3).

**Deeper networks, normalization layers and other architectures (R4)** We experimented with a variety of normalization layers. We note that the sine activation has normalizing properties, and the proposed initialization scheme guarantees standard normal distributed activations at initialization (see suppl.). Likely as a result of this, normalization layers did not lead to performance gains. Similarly, we trained SIRENs with 5, 10 and 15 layers for solving Poisson's equation (learning rate=1e-6). The attached figure shows number of iterations vs. image L2 loss. Deeper networks lead to faster convergence & better final performance, but also introduce noise in the training process, which, however, does not impact stability or convergence (see difficulty of training below). Lastly, we tested sine activations in a set encoder architecture (suppl. Tab. 4). All those new experiments warrant exciting future work and we will add them in the suppl.

**Comment on generalization experiments (R1)** Generalizing across the space of SIRENs is an exciting avenue for future work and we will comment on this in the discussion section. We chose the CelebA image completion task because those experiments in Conditional Neural Processes (Garnelo et al., 2018) seem to be an appropriate benchmark of encoder-based generalization across a space of functions.

**Intuition for sine activation & relation to spectral domain processing (R4)** Our motivation to use sine activations lies in their infinite VC dimension, their non-locality and shift-invariance arising from their periodicity, as well as the promise that they might address the low-frequency bias of ReLU MLPs. Those comments will be added in the paper. **(R1,2,3)** suggest to comment on the connections with spectral domain processing. We first note that a SIREN with a *single hidden layer* can be seen as performing a frequency decomposition as in Fourier Series. However, the function parameterized by a multi-layer SIREN cannot be trivially identified with such a series. SIRENs, like other deep nets, build on *non-linearities*, while spectral methods for PDEs rely on using *linear* decomposition on Fourier Bases. Hence, relationships to spectral methods are not obvious. **(R3)** suggests transforming the inputs into the spectral domain: this is related to the approach taken by PEs [5], which we benchmark.

**Comment on Poisson reconstruction (R1)** pointed out that Poisson equation can be used for surface reconstruction from normals. This is correct and our SDF loss contains $|\nabla \Phi_x - \vec{n}|$; If this was the only term, it would exactly solve a Poisson equation but our loss has additional constraints to solve for functions also satisfying an Eikonal equation. We chose to demonstrate the Poisson reconstruction and editing on images since they are minimal yet useful examples.

**Clarify boundary conditions (R1)** We use different types of boundary values (Neumann, Dirichlet, and mixed BVs) throughout the paper. In the Helmholtz equation, the BVs appear via the PML formulation (energy at infinity is 0), in the SDF they appear as the points on the surface, which must have a value of 0, the gradient/Laplacian experiments do not have BVs expressed on the function itself (Dirichlet) but on its derivatives (Neumann), hence their solution is up to a constant (resp. an additional ramp). This will be clarified in the paper as well as in the suppl.

**Comment on difficulty of training (R4)** SIREN is easy to train and converges quickly and reliably for the applications we demonstrated in the paper. When using a high learning rate with ADAM, some spikes in the error curve typically appear during training (see above figure). However, these do not impact learning, the loss recovers immediately, and the error systematically decreases. We will comment on this aspect in the supplemental.

**Clarify the training with derivatives (R4)** Training using the derivatives (w.r.t inputs) of the model is performed via automatic differentiation (autodiff). Since the inputs to our networks are the coordinates $x$ and the output is the function value $\Phi(x)$, autodiff can be used to evaluate the derivative $\partial \Phi / \partial x$ by computing the gradient of the network's output w.r.t to its inputs. This gradient calculation is automatically added to the computational graph with autodiff, enabling the optimization of the weights during training. This will be stated in the paper and expanded in the suppl.

**Clarify input/outputs (R4)** We will clearly state the input/output pairs for all the experiments in the paper and will include a summary table in the suppl. Reviewers can also refer to the suppl. video describing the setting (input/outputs and the loss formulation) for each experiment.

[Meta-Review · NeurIPS 2020]

All four reviewers note that the paper proposes and investigates an important and very useful class of deep architectures and acknowledge a wide range of potential applications and a number of interesting future extensions. Based on the strongly positive feedback, the paper is recommended to be accepted.